# REPRESENTATION-BASED EXPLORATION FOR LANGUAGE MODELS: FROM TEST-TIME TO POST-TRAINING

**Jens Tuyls**[1,*]   **Dylan J. Foster**[2]   **Akshay Krishnamurthy**[2]   **Jordan T. Ash**[2]
[1]Princeton University   [2]Microsoft Research NYC
jtuyls@cs.princeton.edu, {dylanfoster,akshaykr,ash.jordan}@microsoft.com

## ABSTRACT

Reinforcement learning (RL) promises to expand the capabilities of language models, but it is unclear if current RL techniques promote the discovery of novel behaviors, or simply sharpen those already present in the base model. In this paper, we investigate the value of deliberate exploration—explicitly incentivizing the model to discover novel and diverse behaviors—and aim to understand how the knowledge in pre-trained models can guide this search. Our main finding is that exploration with a simple, principled, **representation-based** bonus derived from the pre-trained language model's hidden states significantly improves diversity and pass@k rates—both for post-training, and in a novel inference-time scaling setting we introduce.

1. For inference-time, exploration with representation-based diversity improves efficiency, consistently improving pass@k rates across a variety of models and reasoning tasks. For example, for `Qwen-2.5-14b-Instruct` we obtain over 50% improvement in verifier efficiency on almost all tasks.

2. For post-training, we show that integrating this exploration strategy into an RL pipeline improves reasoning performance over that of the initial model and over standard RL post-training. For example, on `AIME 2024`, our post-trained `Qwen-2.5-7b-Instruct`'s pass@80 matches the pass@256 of GRPO on the same model, demonstrating a 3x improvement in test-time sample efficiency.

Overall, our findings suggest that deliberate exploration—with the right notion of diversity—is a practical path toward discovery of new behaviors beyond sharpening.[1]

## 1 INTRODUCTION

Reinforcement learning (RL) promises to endow agents with the ability to discover valuable behaviors autonomously, via closed-loop trial and error. For language modeling tasks with verifiable rewards, such as mathematical reasoning and code generation, post-training with reinforcement learning has already enabled impressive breakthroughs (DeepSeek-AI, 2025; OpenAI, 2024). Still, it is unclear whether contemporary RL implementations for language models attain the full promise of reinforcement learning. Rather than unlocking capabilities not present in the pre-trained model, there is increasing evidence (Yue et al., 2025; Gandhi et al., 2025) that existing RL recipes (Schulman et al., 2017; Rafailov et al., 2023; DeepSeek-AI, 2025) may simply amplify or *sharpen* (Huang et al., 2025) behaviors that the base model can already execute, albeit with modest probability. While this can be mitigated through deliberate data curation and some algorithmic interventions (He et al., 2025; Liu et al., 2025; Setlur et al., 2025), data scale and quality are rapidly becoming bottlenecks, particularly in complex, open-ended domains where existing interventions fall short of eliciting desired behavior.

We argue that deliberate exploration—incentivizing the model to discover truly novel and diverse behavior—is an essential ingredient in realizing the full potential of RL for language model reasoning. Exploration has a rich history in both the theory and practice of RL, and exploration techniques tailored to deep networks (Tang et al., 2017; Pathak et al., 2017; Burda et al., 2018; Osband et al., 2019) have received extensive investigation in the context of embodied decision making, including game playing and robotic control. These algorithms proceed from scratch, without pre-training,

---

[*]Work partially completed during an internship at Microsoft Research.
[1]Website and code: `https://rep-exp.github.io`

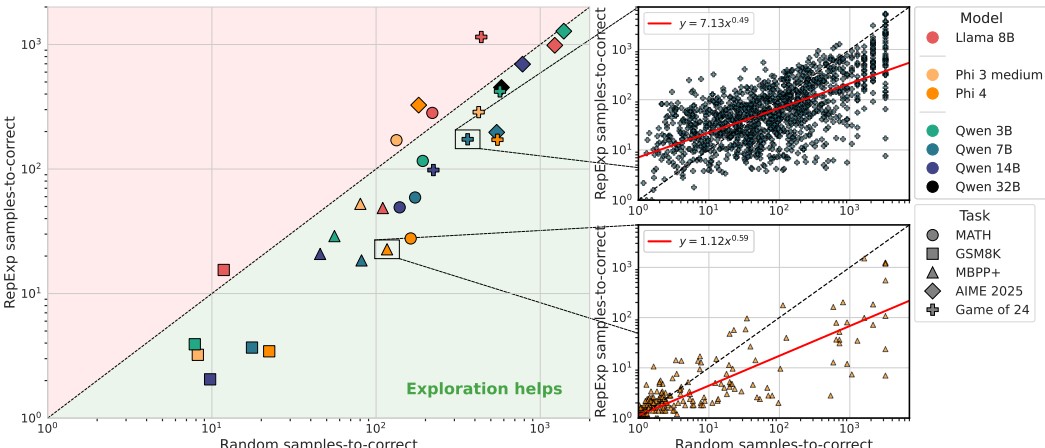

Figure 1: **Representation-based inference-time exploration improves verifier efficiency.** *(Left)* We plot the *samples-to-correct*, the average number of samples until a correct response is selected, for a wide range of tasks and models. We compare two inference-time exploration methods: representation-based exploration (Section 3) and naive (random) sampling from the base model. *(Right)* We display samples-to-correct, disaggregated to each question in the dataset, for two model-task pairs. **We find representation-based exploration improves over random sampling for most model-task pairs.** For example, for `Qwen-2.5-14b-Instruct` we obtain over 50% improvement in verifier efficiency on GSM8K, MATH, MBPP+, and `Game-of-24`. See Section 4.1 for details.

yet rapidly learn complex behaviors, demonstrating that they enable learning beyond the sharpening regime. If we can equip language models with exploration in a similar fashion, we may be able to advance reasoning capabilities without incurring exorbitant data curation costs.

In spite of the potential benefits of exploration, it is unclear which, if any, exploration technique from deep RL can be scaled to modern language models. A central challenge involves the scalable quantification of novelty and behavior diversity—and acting on this information—when the decision space under consideration is the combinatorially large space of language. At the same time, pre-trained language models contain tremendous prior knowledge compared to policies found in traditional embodied settings, which may be the key to guiding efficient exploration. This leads us to ask:

> 1. Can the knowledge in pre-trained representations guide the search for novel behaviors?
>
> 2. Does deliberate exploration have the potential to move beyond sharpening the base model?

## 1.1 CONTRIBUTIONS

Toward answering these questions, we focus on understanding whether exploration with *diversity bonuses* $\texttt{div}(x, y)$ derived from a language model can effectively guide the search for diverse behaviors. We adopt a novel methodology (Section 2) in which we first evaluate exploration in a simple, purely inference-time setting, then integrate our findings into post-training.

**The inference-time selection problem (Section 2).** In this setting, we aim to select a small set of responses $y_1, \ldots, y_k$ from a large set of candidates $y_1, \ldots, y_N$ for a given prompt $x$, such that the chosen set is as diverse as possible, and has high probability of including a positive response. This simple regime allows us to disentangle the role of diversity $\texttt{div}(x, y)$ from other complex RL mechanisms, such as optimization and generalization.

**Representation-based exploration improves diversity and efficiency.** Our main finding is that exploration with a **representation-based** bonus (Section 3) derived from the pre-trained language model's hidden states significantly improves diversity and pass@k rates—both for our inference-time setting and for post-training. Our specific findings are as follows:

1. **Inference-time (Section 4).** Inference-time exploration with representation-based diversity improves verifier efficiency. For example, we obtain over 50% improvement in verifier efficiency

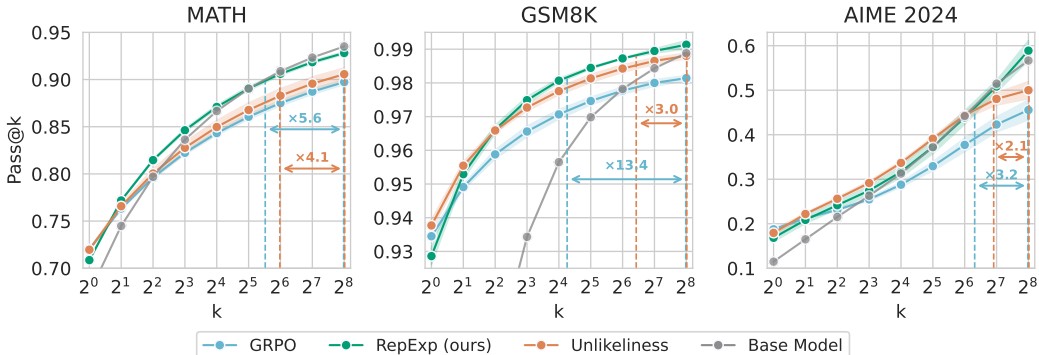

Figure 2: **Pass@k for RL post-training with exploration.** We find that RL generally increases the pass@k for small values of $k$ compared to the base model, but that exploration is required to improve or even preserve base model pass rates for larger values of $k$. For MATH and GSM8K, RepExp roughly matches or improves upon Unlikeliness for $k \geq 2$. For AIME 2024, RepExp is slightly worse than Unlikeliness until $k = 64$, after which it surpasses Unlikeliness for all larger values of $k$. Shaded areas indicate one standard error. Horizontal arrows indicate the test-time sample efficiency improvement for pass@256 of RepExp over GRPO (blue) and Unlikeliness (orange). **RepExp is 2.1-4.1x more sample-efficient than Unlikeliness and 3.2-13.4x more sample-efficient than GRPO.**

over standard sampling for Qwen-2.5-14b-Instruct on GSM8K, MATH, MBPP+ and Game-of-24. See Figure 1 for an overview of our results for this setting.

2. **Post-training (Section 5).** Representation-based exploration can be incorporated into RL post-training, where its pass@$k$ performance is competitive with both GRPO and the base model *uniformly for all $k$* (Figure 2). Notably, **representation-based exploration completely eliminates the "diversity collapse" phenomenon where RL degrades pass@$k$ with respect to the base model for large $k$** (Dang et al., 2025; Yue et al., 2025; Wu et al., 2025). In addition, representation-based exploration induces responses that look much more novel under the base model (Figure 8).

Our findings, particularly these last two points, suggest that deliberate exploration is a practical path toward discovery of new behaviors beyond sharpening. Although our experiments focus on arguably the simplest principled representation-based exploration scheme—for which we already see substantial performance improvements—we expect that our two-pronged evaluation approach will enable a deeper understanding of the benefits and tradeoffs of more sophisticated strategies, which may help realize the full potential of reinforcement learning for language model reasoning.

**Diversity-guided generation (Section 4.2).** As a proof of concept, we also evaluate an inference-time exploration algorithm that uses representation-based diversity to encourage exploration *during the autoregressive generation process itself*. We find that this improves pass@k for large k over naive sampling for Qwen-2.5-7b-Instruct on MATH.

## 2 PROBLEM SETUP: FROM INFERENCE-TIME TO POST-TRAINING

In this section, we describe the two problem settings we consider for exploration: inference-time selection and RL post-training. In what follows, $\pi$ denotes a language model that maps a prompt $x \in \mathcal{X}$ to a distribution over responses $y \in \mathcal{Y}$, and $r^\star(x, y) \in \{0, 1\}$ denotes a verifiable reward function that measures correctness at a task of interest, such as whether the answer to a math question is correct, or whether a Python program passes unit tests.

**Methodology and motivation.** The goal of RL post-training is to find a policy $\pi$ that maximizes the expected reward $\mathbb{E}_{y \sim \pi(\cdot|x)}[r^\star(x, y)]$. Given a budget $k$ of verifier queries per question at each data collection round, post-training algorithms such as GRPO (Shao et al., 2024) update the model iteratively, where in each iteration they sample $k$ responses $y_1, \ldots, y_k \overset{\text{i.i.d.}}{\sim} \pi(\cdot \mid x)$ per prompt $x$ from the current model $\pi$, query the verifier for a reward $r^\star(x, y_i)$ for each response, and use observed rewards to update the model for the next iteration.

If the initial model $\pi$ has poor support over rewarding behavior—i.e., if $r^\star(x, y) = 0$, with high probability under $y \sim \pi(\cdot \mid x)$—common RL algorithms such GRPO or PPO (Schulman et al., 2017)

will not make any progress. This motivates interventions for exploration such as bonuses (Tang et al., 2017; Pathak et al., 2017; Burda et al., 2018; Osband et al., 2019) and alternative sampling strategies (Holtzman et al., 2020; Minh et al., 2025). However, understanding the benefits and tradeoffs of these interventions in RL post-training is challenging because exploration interacts with optimization and generalization. To isolate exploration from these other considerations, we center our investigation around a task we refer to as **inference-time selection**, validating interventions in this setting before integrating them into the RL post-training pipeline.

## 2.1 INFERENCE-TIME SELECTION

In the inference-time selection problem, we aim to use a fixed model $\pi$ to build a set of $k$ responses to a given prompt $x$ that are maximally diverse and have high probability of containing a positive response. As a simple baseline, we may independently sample $k$ responses from the model—potentially with high-temperature sampling, nucleus or min-p sampling, or other modified sampling schemes. However, the limitations of these baselines are (i) they may not effectively capture the model's understanding of diversity, and (ii) by sampling independently, we may waste verifier queries on redundant responses.

Instead, we focus on *selection-based* approaches that initially sample a large set of candidate responses to the prompt, then use a diversity bonus $\text{div}(x, y)$ derived from the model to filter this set down to a smaller, more diverse "coreset" (Clarkson, 2010; Feldman et al., 2020) that is passed to the verifier. Formally, we consider the following protocol: For each prompt $x$, we (1) sample an initial batch of $N$ responses $y_1, \ldots, y_N \sim \pi(\cdot \mid x)$, (2) use the inference-time selection algorithm $\text{Alg}$ to select $k$ of these responses (a subset $S \subset [N]$ of size $|S| = k$), and (3) query the verifier and record if any of the selected responses are rewarding. That is, we measure pass@$k$,

$$\mathbb{E}_{y_1, \ldots, y_N \sim \pi(\cdot | x)} \Big[ \mathbb{E}_{S \sim \text{Alg}(x, y_1, \ldots, y_N)} \Big[ \max_{i \in S} [r^\star(x, y_i)] \Big] \Big]. \tag{1}$$

Importantly, the filtering algorithm operates without the verifier, and so successfully retaining high-quality responses translates to improved *verifier efficiency* (i.e., number of responses for which we query the verifier) over the initial set of responses. Thus, a useful diversity bonus $\text{div}(x, y)$ should yield a coreset that is maximally "exploratory," in the sense that it is the most diverse set of responses that can be selected for a fixed budget of verifier queries. For example, in math reasoning settings, we would like to select the distinct-but-plausible proof strategies for a given problem, thus covering the space of potential proofs and maximizing the chance of selecting a correct one.

**Remark 2.1.** *While we mainly introduce inference-time selection as a stepping stone to post-training (i.e., algorithms in this setting are not more compute-efficient than naive sampling, even if they are more verifier-efficient), we do expect inference-time exploration to be useful in its own right for domains where querying a verifier is costly or difficult (e.g., collecting feedback from expert-level annotators), allowing for more sample- and hence cost-efficient data curation. For preliminary results in one such domain, please refer to Appendix D.*

## 2.2 REINFORCEMENT LEARNING POST-TRAINING

As described earlier, RL post-training (e.g., with GRPO or PPO) proceeds by iteratively sampling batches of responses, querying the verifier, and using the feedback to update the current policy. After selecting a checkpoint $\hat{\pi}$, we evaluate performance via pass@$k$ under standard generation, $\mathbb{E}_{y_1, \ldots, y_k \sim \hat{\pi}(\cdot | x)} \big[ \max_{i \in [k]} r^\star(x, y_i) \big]$. There are two natural approaches to integrate exploration methods into this process. The first is to adjust the independent sampling process only (e.g., through nucleus sampling or min-p sampling), and the second is to augment the training objective with an exploration bonus $\text{div}(x, y)$. Our experiments focus on the latter approach; however, based on our results for inference-time selection, we expect that incorporating representation-based exploration into the sampling process will also improve RL post-training performance. Indeed, our two-pronged evaluation is motivated by the hypothesis that *diversity bonuses $\text{div}(x, y)$ that perform well at inference-time also perform well in post-training.*

## 3 REPRESENTATION-BASED EXPLORATION: INFERENCE-TIME AND RL

Having motivated our setup, we now turn to the question of what diversity bonuses $\text{div}(x, y)$ are suitable for exploration with language models. While many metrics have been proposed in the literature (Tang et al., 2017; Pathak et al., 2017; Burda et al., 2018; Osband et al., 2019), the challenge in adapting these techniques to language models is to simultaneously (i) capture the

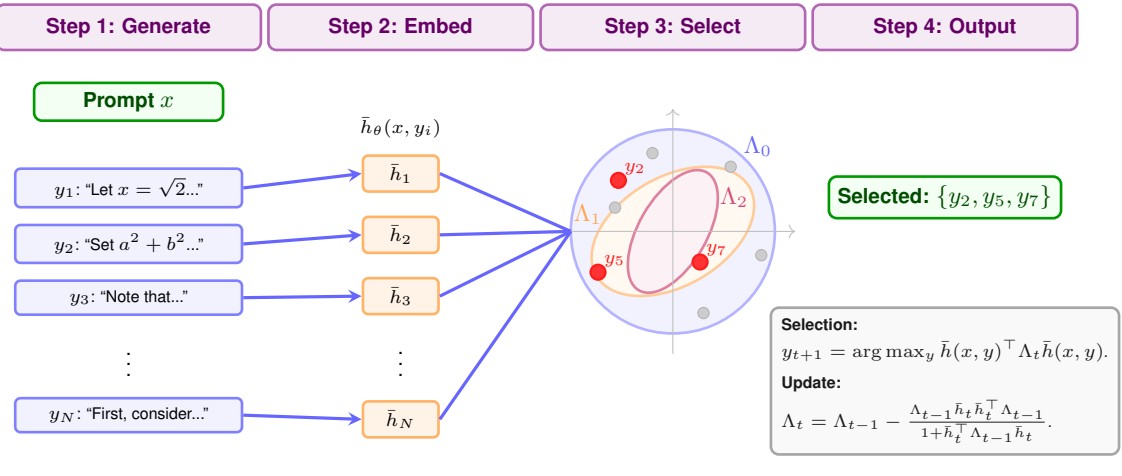

Figure 3: **RepExp for inference-time exploration.** Given a prompt, RepExp selects a diverse set of responses from a large pool by optimizing elliptical bonuses computed using representations from the language model.

model's understanding and (ii) allow for efficient computation at scale. For example, count-based exploration (Tang et al., 2017) is simple, but unsuited to large decision spaces. On the other hand, approaches based on intrinsic curiosity (Pathak et al., 2017), random network distillation (Burda et al., 2018), and posterior sampling (Osband et al., 2019) are better suited to large or continuous spaces, but require additional learning machinery (i.e., auxiliary networks), which introduces significant complexity when scaling to language models.

We focus our experiments on an exploration strategy that avoids these shortcomings: An adaptation of elliptic bonuses and sampling—a de facto standard for linear bandits and active learning (Abbasi-Yadkori et al., 2011; Chu et al., 2011; Ash et al., 2021; Henaff et al., 2022; Saran et al., 2023; Foster et al., 2025)[2]—with a representation derived from the language model's hidden states. This approach is arguably the simplest principled strategy that is appropriate for language models, and already yields significant performance improvements in our experiments. Our use of elliptic bonuses is particularly inspired by Foster et al. (2025), who prove that test-time exploration with such bonuses has provable computational benefits in a simplified language model setting with frozen features.

At a high level, elliptical bonus methods operate over a $d$-dimensional feature space and adopt a linear-algebraic measure of novelty: given previously seen feature vectors $h_1, \ldots, h_{i-1}$ the novelty (or bonus) of a new feature vector $h$ is defined as

$$\mathtt{div}(h \mid h_{1:i-1}) = h^\top \Sigma_i^{-1} h \qquad \Sigma_i = \lambda I_d + \sum_{j<i} h_j h_j^\top \tag{2}$$

These bonuses are grounded in the theory of linear regression: If we fit a linear model $f_\theta(h) = \langle \theta, h \rangle$ on features $h_1, \ldots, h_{i-1}$ (with associated regression targets), the prediction error on $h$ will be bounded by $\mathtt{div}(h \mid h_{1:i-1})$ (Lattimore & Szepesvári, 2020). Thus, $\mathtt{div}(h \mid h_{1:i-1})$ reflects novelty, as it will be large for features $h$ that are poorly represented by the training dataset.

To adapt elliptical bonuses to language models, we use representations extracted from the model itself as the feature vectors. Formally, given a prompt $x$ and a response $y_i = y_i^1, \ldots, y_i^T$ of $T$ tokens, we form the feature vector as $\bar{h}_\theta(x, y_i) := \frac{1}{T} \sum_{t=1}^{T} h_\theta(x, y_i^{1:t})$ where $h_\theta(x, y_i^{1:t}) \in \mathbb{R}^d$ is the last-layer hidden state of the model on input $(x, y_i^{1:t})$ (the activation prior to the unembedding matrix).

---

[2]Indeed, elliptical bonuses are ubiquitous in linear bandits and reinforcement learning, the simplest non-tabular RL setting, where they have strong provable guarantees. Beyond this, elliptic bonuses and iterative schemes such as Algorithm 1 have a long history in the theory of optimal experimental design (Kiefer & Wolfowitz, 1960; Pukelsheim, 2006; Allen-Zhu et al., 2021) and active learning (Cesa-Bianchi et al., 2009; Agarwal, 2013; Gu et al., 2014; Chaudhuri et al., 2015)

**Algorithm 1** RepExp

1: **input:** Embeddings $\bar{h}_\theta$ (abbrv. $\bar{h}$),
   generations $\mathcal{Y}$ for prompt $x$,
   budget $k$, regularization param. $\lambda$.
2: Initialize $L \leftarrow \{y_1\}, y_1 \sim \text{Unif}(\mathcal{Y})$.
3: Initialize inverse covariance $\Lambda_0 = \lambda^{-1} I_d$.

4: **for** $t = 1$ to $k - 1$ **do**
5: $\quad \Lambda_t \leftarrow \Lambda_{t-1} - \frac{\Lambda_{t-1} \bar{h}(x,y_t) \bar{h}(x,y_t)^\top \Lambda_{t-1}}{1 + \bar{h}(x,y_t)^\top \Lambda_{t-1} \bar{h}(x,y_t)}$.

6: $\quad y_{t+1} = \underset{y \in \mathcal{Y}}{\arg\max} \, \bar{h}(x,y)^\top \Lambda_t \, \bar{h}(x,y)$.

7: $\quad L \leftarrow L \cup \{y_{t+1}\}$.
8: **return:** $L$.

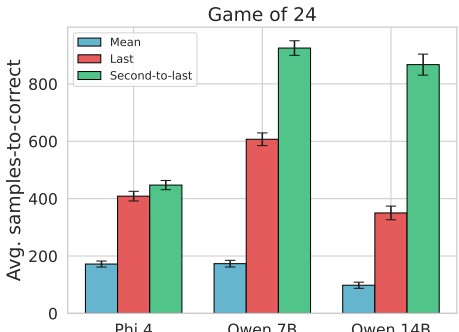

Figure 4: **Representation ablation.** We compare averaging all token representations to using those from the penultimate or final token. **Averaging is over 2x more sample efficient.**

In Figure 4, we ablate this choice by comparing it to the effectiveness of using representations at the last token $h_\theta(x, y_i^{1:T})$ or penultimate token $h_\theta(x, y_i^{1:T-1})$ instead. We reduce dimensionality to 512 using a sparse random projection (Li et al., 2006); see Appendix B for details.

**Representation-based exploration for inference-time selection.** Figure 3 presents RepExp, our main algorithm for inference-time selection using representation-based elliptical bonuses. Here, given a single prompt $x$ and a set of candidate generations $\mathcal{Y} = \{y_1, \ldots, y_N\}$, we iteratively select the generation that maximizes the elliptical bonus via $y_{t+1} = \arg\max_{y \in \mathcal{Y}} \bar{h}_\theta(x, y) \Sigma_t^{-1} \bar{h}_\theta(x, y)$, leveraging the representations $\bar{h}_\theta(x, y)$ described above. We efficiently update the inverse covariance matrix $\Sigma^{-1}$ using the Woodbury identity for $O(d^2)$ time per step (Vetterling & Press, 1992). We formally present our procedure in Algorithm 1.

**Representation-based exploration for RL post-training.** For our post-training experiments, we use the same representations $\bar{h}_\theta(x, y)$ as above, but directly augment the rewards with elliptic bonuses instead of performing coreset selection. Concretely, given the current iterate $\pi_\theta$ in GRPO, we first sample a group of $k$ responses $y_1, \ldots, y_k \overset{\text{i.i.d.}}{\sim} \pi_\theta(\cdot \mid x)$ for each prompt $x$. Letting $\Sigma := \lambda I_d + \sum_{i=1}^k \bar{h}_\theta(x, y_i) \bar{h}_\theta(x, y_i)^\top$, we define the reward for response $y_i$ as[3] $r^\star(x, y_i) + \beta \cdot \bar{h}_\theta(x, y_i)^\top \Sigma^{-1} \bar{h}_\theta(x, y_i)$, where $\beta > 0$ is a bonus parameter. While one could also imagine performing inference-time coreset selection in the loop with GRPO, this approach is more practical and efficient, and it achieves significant improvements in performance. We refer the reader to Section 5 and Appendix C for further details.

**Why representation-based elliptical bonuses?** We summarize several desirable properties of these bonuses. First, by leveraging the hidden state of the model in featurization, the bonuses capture rich information about the generations, thereby incorporating the language model's prior knowledge. Second, the method is *history-aware*[4]: the covariance matrix summarizes all previously selected generations, and redundancy with previous selection (in representation space) is penalized. Finally, the method is simple and scalable, involving no additional learning machinery and using rank-one updates to avoid costly matrix inversions.

## 4 INFERENCE-TIME EXPLORATION: EXPERIMENTAL RESULTS

In this section, we investigate the performance of representation-based exploration for the inference-time selection problem. We detail the experimental setup in Section 4, present main findings in Section 4.1, and present additional experiments with a "token-level" variant in Section 4.2.

**Datasets.** We use the *test* splits of the following five datasets: MATH (Hendrycks et al., 2021), GSM8K (Cobbe et al., 2021), MBPP+ (Liu et al., 2023), Game-of-24 (Yao et al., 2023), and AIME 2025. We chose these tasks as they cover easy (GSM8K), medium (MATH), and harder (Game-of-24, AIME)

---

[3]The bonus here can be interpreted as a *leverage score* for $y_i$ (Drineas et al., 2006; Cohen et al., 2015).

[4]In the RL setting, we do not let the covariance matrix persist across multiple iterations of the same question, and hence there it is only *group-aware*.

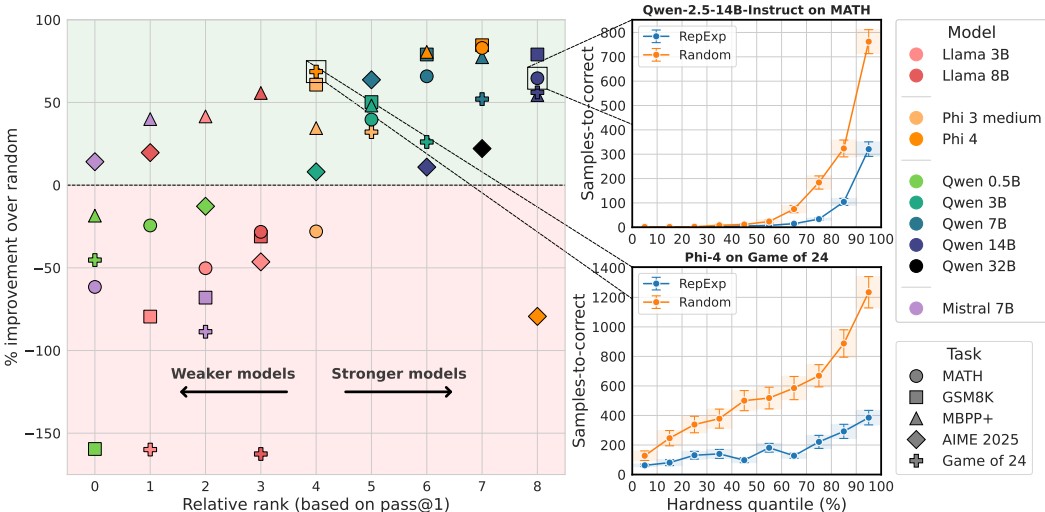

Figure 5: **A closer look into when RepExp provides improvement.** *(Left)* For each task, we rank models according to their pass@1 rate (the weakest model has rank 0, and the strongest has rank 8). We then plot relative improvement (%) of RepExp over random sampling, sorting by rank on the x-axis. While RepExp can hurt weaker models (e.g., Qwen-2.5-0.5B-Instruct), we find **stronger models almost always benefit from exploration** (e.g., Qwen-2.5-14B-Instruct). *(Right)* For two different model-task pairs, we plot the samples-to-correct as a function of *question hardness*. Hardness is measured by the samples-to-correct from a high-quality third-party model (GPT-4o mini). We find that **RepExp has the greatest benefit on harder examples** (e.g., the hardest 20% of questions on MATH). Shaded areas indicate one standard error.

difficulty levels in math. In addition, we include MBPP+ to verify that our findings transfer to the coding domain. For a more detailed overview of these datasets, please refer to Appendix B.1.

**Models.** We consider a range of model families and sizes: Phi-3-Medium (Abdin et al., 2024a) and Phi-4, Llama-3.2-3B-Instruct and Llama-3.1-8B-Instruct (Dubey et al., 2024), (Abdin et al., 2024b), Qwen-2.5-X-Instruct (Qwen et al., 2024) for X $\in$ {0.5B, 3B, 7B, 14B, 32B}, and Mistral-7B (Jiang et al., 2023).

**Algorithms.** In our experiment protocol, we initially draw a pool of $N$ candidate generations from the base model, where unless otherwise specified we use temperature $\tau = 1.0$ and top-p $= 1.0$, which we refer to as vanilla settings (for MBPP+, we set top-p $= 0.95$). Then we compare RepExp with budget $k$ with the baseline of random sampling (without replacement) of $k$ responses from this pool. We consider generating the pool using different samplers such as nucleus and min-p sampling in Figure 6, but always use random sampling without replacement as the baseline. See Appendix B.1 for further details.

## 4.1 BENEFITS OF REPRESENTATION-BASED EXPLORATION

We present our results as a series of Research Findings (RF), expanding on the findings in Figure 1.

**RF1: RepExp improves verifier efficiency across models and tasks.** In Figure 1, we plot the *samples-to-correct*, defined as the expected number of samples $k$ with which we query the verifier before finding a correct answer, for all model-task pairs. We compare RepExp, which picks responses to a fixed question according to Algorithm 1, with the random sampling baseline. For both algorithms, we average the samples-to-correct across all questions in the dataset. Our results show the bulk of the data fall below the line $y = x$, indicating exploration improves over random sampling in most cases. For example, we find RepExp obtains a 50% improvement in samples-to-correct for Qwen-2.5-14b-Instruct in MATH, GSM8K, MBPP+, and Game-of-24.

**RF2: The benefits of RepExp grow with model strength.** Since RepExp relies on the model's internal representations, it is natural to hypothesize that weaker models might have worse representations and thus benefit less from exploration. To validate this hypothesis, we expand the collection of models in Figure 1 to include additional weaker models (e.g., Qwen-2.5-0.5B-Instruct and

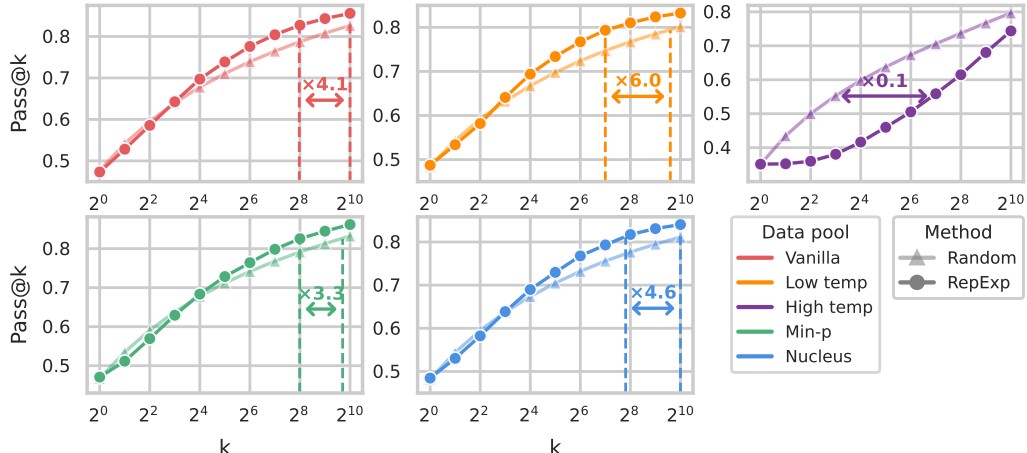

Figure 6: **Benefits of RepExp across data pools**, for the inference-time exploration setup in Figure 1 (Section 4.1). We plot the pass@$k$ curve for random vs. RepExp across five different data pools (base samplers). **RepExp on top of vanilla generation outperforms random sampling on top of *any* of the generation strategies.** Moreover, except for the high-temperature pool, RepExp over a pool improves verifier efficiency, with **3x to 6x improvement over random sampling** for that pool.

Mistral-7b). For each task, we rank models according to their pass@1 performance, and plot the relative improvement of representation-based exploration over random sampling in Figure 5. We indeed observe a strong correlation between model strength and the benefit from representation-based exploration: weaker models (e.g., Qwen-2.5-0.5B) experience no benefit or even degradation, while the strongest models (e.g., Qwen-2.5-32B) almost uniformly benefit.

**RF3: RepExp provides more improvement for harder questions.** Beyond **RF2**—which provides insight into the benefits of RepExp across *models*—we also evaluate the benefits across *question difficulty*, for a fixed model and task. To this end, we sort all questions for a given task by their samples-to-correct under random sampling with a reference model (GPT-4o-mini). We then group the questions in bins, each containing 10% of the dataset, and plot the average samples-to-correct for each bin for both RepExp and random sampling. As displayed in Figure 5 (right), RepExp matches or improves verifier efficiency across all bins, with the largest improvements on the hardest bins (e.g., the hardest 20% of questions on MATH). Concretely, on the hardest Game-of-24 questions, we find that RepExp with Phi-4 provides a 3x improvement in verifier efficiency.

**RF4: RepExp improves verifier efficiency over standard generation modifications.** We now investigate the effect of alternative base sampling strategies that might already induce diversity. Using Qwen-2.5-7B-Instruct on MATH, we change the underlying generation strategy to use one of five different generation settings: vanilla (no changes), low temperature ($\tau = 0.6$), high temperature ($\tau = 1.5$), min-p (Minh et al., 2025) ($\tau = 1.5, p = 0.05$), and nucleus sampling (Holtzman et al., 2020) (top-p $= 0.9$). In Figure 6, we find that RepExp improves verifier efficiency in all settings, except for when paired with high-temperature sampling. We suspect this is because high-temperature sampling tends to produce less coherent responses, which may look novel in representation space, yet do not necessarily contain correct answers.

## 4.2 Extension: Representation-based exploration at the token level

While useful in its own right as a testbed for benchmarking the viability of exploration methods, one drawback of the inference-time selection setting is that compute—as measured by $N$, the size of the per-question data pool—may need to be rather large relative to $k$ for selection to yield improvements. As an extension, we conduct a preliminary investigation into algorithms that use elliptic bonuses to guide the autoregressive generation process itself, removing the need to generate such a pool at all.

**Representation-based exploration for autoregressive generation.** To guide sampling for improved diversity, given a budget $k$, we use features from responses $1, \ldots, i - 1$ to guide the generation of the $i$th response by modifying the logits at every generation step. Specifically, consider the $i$th generation for a given prompt $x$. At each position $t$ *within* the generation, we perturb the

$|V|$-dimensional token-level logit vector as:

$$\tilde{\mathbf{z}}^{(i)}(x, y_{<t}) = \mathbf{z}^{(i)}(x, y_{<t}) + \beta \cdot \mathbf{b}^{(i)}(x, y_{<t}, V), \quad \tilde{\mathbf{z}}^{(i)}(x) \in \mathbb{R}^{|V|},$$

where the bonus $\mathbf{b}^{(i)}(x, y_{<t}, V)$ is a token-level elliptic bonus, defined as:

$$\mathbf{b}_j^{(i)}(x, y_{<t}, V) = \sqrt{\tilde{h}_\theta(x, y_{<t}, v_j)^\top \Sigma_{(i)}^{-1} \tilde{h}_\theta(x, y_{<t}, v_j)},$$

for $v_j \in V$. Here $\tilde{h}_\theta(x, y_{<t}, v_j)$ is a mean-centered Transformer representation for sequence $(y_{<t}, v_j)$; see Appendix B.2 for further details.

### RF5: RepExp for autoregressive generation improves solve rate.

In Figure 7, we visualize the effect of token-level representation-based exploration for `Qwen-2.5-7B-Instruct` on the `MATH` task. We use two values for the bonus parameter $\beta$ (0.5, 1.0) and compare the pass@k to vanilla autoregressive generation for the 200 hardest (but solvable) questions in `MATH`, as judged by `GPT-4o-mini`. While token-level exploration tends to solve fewer problems compared to vanilla generation when given a small budget, this trend reverses when the budget exceeds $512-640$ (depending on choice for $\beta$). Figure 9 further shows that the improvement in solve rate is largest on the hardest questions. These results are encouraging, though further research is required (e.g., on more tasks and models) before one can draw a definitive conclusion. Further, our implementation is not optimized for efficiency and requires at least one additional forward pass per generation step compared to naive sampling.

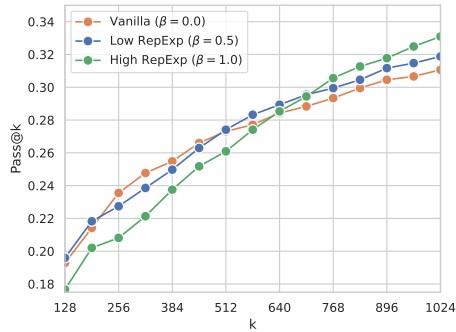

Figure 7: **Representation-based exploration at the token level**, compared to naive autoregressive generation ($\beta = 0$) for inference-time exploration. RepExp improves pass@k for large $k$ over naive sampling.

## 5 EXPLORATION FOR RL POST-TRAINING

Following the methodology in Section 2, we now investigate the use of representation-based exploration to guide the RL post-training process.

**Tasks and models.** We use `Qwen-2.5-7b-Instruct` evaluated on `MATH`, `GSM8K`, and `AIME 2024`. Because there are only 30 questions in `AIME 2024`, we follow Yu et al. (2025) and use the `DAPO-Math-17K` dataset for training, leaving `AIME 2024` for evaluation only. Please refer to Appendix C for exact details on train, validation, and test splits for all tasks.

**Baselines.** We compare our method with three baselines: **(1) Unlikeliness** (He et al., 2025) modifies GRPO by scaling the extrinsic rewards by a value inversely related to the likelihood of a generation under the current policy. **(2) GRPO** is simply an unmodified version of the original GRPO algorithm. **(3) Base Model** is the original untrained model included as a reference point to see if methods can improve upon it, especially at high values of $k$.

**Representation-based Exploration (RepExp).** We augment the rewards in GRPO with representation-based bonuses as described in Section 3. Concretely, we add sequence-level elliptic bonuses to the binary extrinsic rewards provided by the verifier: $r_i = R(x, y_i) + b(x, y_i)$ for the $i$-th rollout $y_i$ of a given prompt $x$. As mentioned in Section 3, we specifically use leverage score-like elliptic bonuses, which allow easier control over their scale as they are bounded in $[0, 1]$. The covariance matrix $\Sigma$ used to compute bonuses is re-initialized for each batch of RL training. This way, the bonus $b(x, y_i)$ measures the novelty of $y_i$ only with respect to the other rollouts in the batch—previously generated sequences for $x$ are not considered. To better maximize the bonus along all relevant directions in representation space, we draw a new random projection of $\bar{h}_\theta(x, y_i)$ at each optimization step. For further details, please refer to Appendix C.

### RF6: RepExp improves pass@k.

Figure 2 compares pass@$k$ curves after training for all methods. In line with earlier work, we find that all instantiations of GRPO improve the pass@k for small values of $k$, and that standard GRPO degrades performance relative to the base model for large values of $k$ (Yue et al., 2025). Exploration appears to be an essential part of mitigating the latter effect: policies fit using RepExp preserve or improve pass@$k$ for large values of $k$ with limited reductions for small $k$. This phenomenon is more pronounced for RepExp than for Unlikeliness.

**RF7: RepExp responses look novel.** To provide further insight into whether RepExp is able to move beyond merely sharpening the base model, we run an additional experiment inspired by Figure 4 in Karan & Du (2025). Specifically, we sample a single response from the base model, the GRPO post-trained model, and the RepExp post-trained model for each question in the full test set of MATH. We then score all responses under the base model in terms of log likelihood. We plot the resulting histogram in Figure 8. We find that responses from RepExp tend to be less likely under the base model, indicating that it generates more novel responses and is therefore not merely performing sharpening. In contrast, notice that standard GRPO exhibits sharpening behavior, as demonstrated by the movement of probability mass towards the right with respect

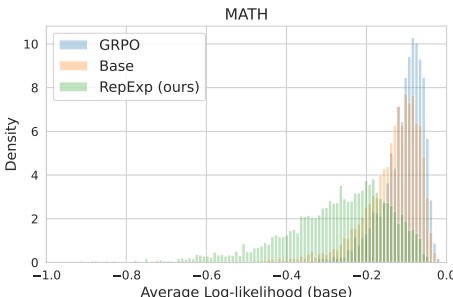

Figure 8: **Anti-sharpening behavior of RepExp.** We plot the histogram of average log-likelihoods evaluated under the base model. While GRPO exhibits sharpening, the responses from RepExp are significantly more novel. See Appendix E for results on GSM8K.

to the base model. Taken together with RF6, these results suggest that with the right exploration strategy, we may be able to escape the sharpening regime and discover novel model behaviors.

## 6 DISCUSSION

**Related work.** Several recent works aim to encourage exploration in language models, either by adapting exploration techniques from deep reinforcement learning, or by augmenting PPO or GRPO in ways that are more specialized to language models (He et al., 2025; Cheng et al., 2025; Chen et al., 2025b; Zhou et al., 2025; Setlur et al., 2025; Liu et al., 2025). Examples of the former include count-based exploration via pseudo-counts (Bai et al., 2025) and at the outcome level (Song et al., 2025), random network distillation (Liu et al., 2024b; Gao et al., 2025), and posterior sampling (Dwaracherla et al., 2024). Examples of the latter include rewarding unlikely-but-correct responses (He et al., 2025), entropy bonuses (Cheng et al., 2025), adapting the number of rollouts based on question difficulty (Yang et al., 2025), training the model to explore in-context (Setlur et al., 2024), using a learned classifier to jointly optimize diversity and quality (Li et al., 2025), adding a diversity term based on determinantal point processes to the RL objective (Chen et al., 2025a), and reformulating post-training to more-directly maximize the pass@$N$ objective (Balashankar et al., 2024; Chow et al., 2025; Chen et al., 2025b; Walder & Karkhanis, 2025; Tang et al., 2025). Among these, we experiment with the unlikeliness reward approach of He et al. (2025) as a baseline, due to its robust performance and clean implementation. More generally, our work is unique in (1) the specific representation-based objective, and (2) our focus on *inference-time* as a means to validate methods with minimal confounding factors. See Appendix A for a detailed overview.

**Final remarks.** Our work shows that deliberate exploration is a viable path toward expanding the reasoning capabilities of language models, offering the possibility of discovering novel behaviors that would be unlikely under naive sampling. While our results show that representation-based diversity is effective at incentivizing exploration, the algorithm design space for exploration techniques is vast, and there is still much to understand regarding how to best use the knowledge encoded in foundation models to guide exploration. Along these lines, natural directions for future work include:

1. *Scaling up RL compute,* and combining exploration with other techniques known to improve reasoning behavior in RL post-training, such as prolonged reinforcement learning (Liu et al., 2025).

2. *Exploration for autoregressive generation.* Our results in Section 4.2 show that incentivizing diversity during autoregressive generation is a promising approach to reducing the computational burden of exploration, but much remains to be done in terms of (1) understanding which diversity metrics are most helpful, and (2) optimizing the implementation to close the compute gap.

3. *Beyond verifiable rewards.* How can we deliberately incentivize exploration in domains without verifiable rewards, while simultaneously mitigating reward hacking?

ACKNOWLEDGEMENTS

We thank Adam Block, Qinghua Liu, and Max Simchowitz for valuable feedback on this work. We thank Manan Tomar, Audrey Huang, Spencer Whitehead, Prabhat Nagarajan, and Karthik Narasimhan for support and encouragement throughout the project.

## REPRODUCIBILITY STATEMENT

To ensure reproducibility of our results, we have listed all relevant details (hyperparameters, experiment resources, etc.) for the inference-time experiments in Appendix B, and those for RL post-training in Appendix C. In addition, the code for all our experiments and plots can be found at `https://rep-exp.github.io`. We used Weights & Biases for experiment tracking and visualizations to develop insights for this paper.

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

## A  ADDITIONAL RELATED WORK

**Exploration at test time.**    Test-time alignment techniques for language models are an active area of research with many complementary threads (Khanov et al., 2024; Chen et al., 2024; Shi et al., 2024a; Liu et al., 2024a; Jinnai et al., 2024; Shi et al., 2024b), but exploration has not typically been the focus of this line of work.

Most closely related to our work, Setlur et al. (2025), propose a test-time exploration approach based on the idea of *learning to explore in-context*. They propose to encourage exploration within a long chain of thought by training the LLM to chain operations such as generation, verification, and refinement together in search of a solution. This is somewhat complementary to our inference-time exploration framework, which aims to improve diversity across parallel generations once the model is fixed; these techniques could potentially be combined.

Also related, Xu et al. (2025) consider the problem of learning from language (non-verifiable) feedback, and propose an iterative prompting approach to enable exploration at test time; their work focuses on simpler exploration domains, but with more difficult, implicit feedback.

**Exploration in RL post-training.**    Exploration in RL post-training for reasoning is a growing area of research, motivated by the observation that standard techniques tend to simply sharpen responses already covered by the base model (Yue et al., 2025; Gandhi et al., 2025; Wu et al., 2025). A number of recent works, discussed below, aim to improve diversity and expand the reasoning frontier by incorporating bonuses into the GRPO objective or by otherwise augmenting it. Briefly, our work is unique in terms of (1) the specific representation-based diversity objective we focus on, and (2) our focus on *inference-time exploration* as a means to validate diversity metrics before applying them to post-training.

He et al. (2025) introduce an *unlikeliness reward* to GRPO, which reweights the reward by ranking generations according to their unlikeliness under the sampling policy. Unlikeliness reward is a form of diversity metric, similar to our representation-based diversity metrics. Cheng et al. (2025) observe that high-entropy (high uncertainty) tokens in the model's output often correspond to critical reasoning steps, and augment the GRPO objective with entropy bonuses to encourage exploration at these high-entropy steps. Entropy can be seen as another form of diversity metric in our setup. Another option is to *learn* the diversity metric as in Li et al. (2025), who use a learned classifier to determine whether a pair of responses is semantically equivalent. They then use the classifier score to scale the reward in GRPO to joinly optimize quality and diversity. Our work, in contrast, does not require training any auxiliary models for computing diversity.

Various works (Balashankar et al., 2024; Chow et al., 2025; Chen et al., 2025b; Walder & Karkhanis, 2025; Tang et al., 2025) formulate the problem of directly post-training to maximize the pass@$N$ objective, deriving approximate gradient estimators and using them for policy optimization. As discussed in Chow et al. (2025); Chen et al. (2025b), these gradient estimators *implicitly* encourage exploration, since they allow the model to distribute probability mass across a more diverse range of responses when it is uncertain about the correct answer. Our work instead focuses on using the language model representations to *deliberately* incentivize novel behaviors, including in a novel inference-time setting.

Zhou et al. (2025) consider a setting where ground truth answers are available (as opposed to just rewards), and propose to encourage exploration by prompting the model to generate self-explanations for the ground truth answers and incorporating this as an SFT term in the GRPO loss. Unlike our method, their approach does not directly optimize for diversity, and cannot be used in settings where ground truth answers are unavailable (e.g. coding).

Yang et al. (2025) investigate the role of "breadth" (batch size) and "depth" (number of rollouts) in RLVR. They show that increasing breadth through full batch updates and increasing depth through more rollouts for harder questions has complementary benefits and overall improves pass@1 and pass@k performance. We view this as orthogonal and potentially complementary to our work.

Lanchantin et al. (2025) introduce diverse preference optimization (DivPO), an alignment method to optimize for both quality and diversity. In contrast to our work, their method is designed for the RLHF setting and applied to non-reasoning tasks (e.g. creative writing).

Concurrent work of Song et al. (2025) adapts tabular UCB-style bonuses to language model post-training with GRPO, but their approach—unlike representation-based exploration—is only suitable

for domains with a small, discrete set of possible outcomes. Other concurrent work of Chen et al. (2025a) optimizes a diversity term along with the rewards, where the diversity term captures the volume of the responses in representation space by computing the determinant of the gram matrix. While related, our method instead adds a leverage-based score to the reward of each individual response.

Lastly, Liu et al. (2025) take a complementary approach and aim to incentivize reasoning beyond the base model through (1) prolonged RL training (increasing the overall amount of training steps), and (2) periodically resetting the reference model; they show that this can increase pass@$N$ performance beyond the base model in a variety of reasoning tasks. This approach is complementary, and could likely be combined with our techniques.

**Representation-based diversity.** Our findings regarding benefits of inference-time exploration with representation-based diversity parallel the findings of Ivison et al. (2025), who evaluated the effectiveness of various data selection schemes for instruction tuning, and found a similar representation-based scheme to be the most effective when normalized for compute. In addition, there is a long line of work relying on representation-based exploration for RL through elliptic bonuses in non-LLM settings (Agarwal et al., 2020a;b; Henaff et al., 2022; Ash et al., 2021).

**Adapting exploration techniques from deep reinforcement learning.** Various papers have adapted exploration techniques from deep learning to language models, including Bai et al. (2025) (count-based exploration), Gao et al. (2025) (random network distillation), and Liu et al. (2024b); Dwaracherla et al. (2024) (posterior sampling).[5] These works show initial promise in terms of sample complexity benefits, but their potential to explore beyond the base model in reasoning domains has not been evaluated to our knowledge. In addition, these methods require additional learning machinery (e.g., auxiliary networks), which introduces significant complexity when scaling to language models.

**Theoretical analysis of language model exploration.** On the theoretical side, our work draws on Foster et al. (2025), who prove that test-time exploration with representation-based diversity has provable computational benefits in a simplified linear setting. Our RepExp algorithm for test-time exploration can be viewed as a simplified, practical adaptation of their theoretical algorithm.

Other theoretical works on exploration with language models include the XPO algorithm of Xie et al. (2024) and related algorithms by Cen et al. (2024); Zhang et al. (2024),[6] which augment the Online DPO objective with exploration bonuses inspired by the optimism principle. To our knowledge, these techniques have only been evaluated on RLHF tasks, and Foster et al. (2025) show that there may be computational barriers to implementing them in a way that is faithful to the theoretical guarantees.

# B    DETAILS FOR INFERENCE-TIME EXPERIMENTS (SECTION 4)

## B.1    DETAILS FROM SECTION 4.1

**Hyperparameters.** In Algorithm 1, we set $\lambda = 1.0$. For all models and tasks, we perform a sparse projection from the respective model hidden dimension to $d = 512$.

**Preprocessing.** In Algorithm 1, after obtaining the representations $\bar{h}_\theta$ for every generation $y$ for a fixed prompt $x$, we sparse project all representations down and then mean-center where the mean is taken across the response-level representations.

**Datasets.** Below we provide a brief overview of all datasets along with relevant numerical details. Note that "vanilla" sampling settings refer to $\tau = 1.0$, top-p $= 1.0$, and min-p $= 0.0$. We also do not use top-k sampling in any of the coreset experiments. Finally, we only use the test split of every dataset for all our inference-time experiments, unless specified otherwise.

- MATH. This dataset contains 12.5k problems from high school math competitions, split into 7.5k training examples and 5k test examples. For each question in the test split, we generate $6400$ responses using vanilla settings and set the maximum response length per generation to $512$ tokens.

---

[5]See also Arumugam & Griffiths (2025), which uses a pre-trained model to simulate posterior sampling in-context for multi-turn sequential decision making tasks.

[6]Cen et al. (2024); Zhang et al. (2024) concurrently proposed similar algorithms to XPO, but did not provide non-trivial theoretical guarantees (e.g., guarantees that indicate benefits over purely passive exploration).

- GSM8K. This dataset contains 8.79k grade school math word problems, split into 7.47k training examples and 1.32k test examples. For each question in the test split, we generate 6400 responses using vanilla settings and set the maximum response length per generation to 512 tokens.

- MBPP+. This dataset contains 378 basic Python programming problems that are a curated subset of the full MBPP (Austin et al., 2021) dataset with more test cases. Since the dataset does not come with any train or test splits, we use the full set of questions for our experiments. For each problem, we generate 6400 responses using vanilla settings, except that we set top-p $= 0.95$. We set the maximum response length per generation to 768 tokens.

- Game of 24. This dataset contains 1.36k questions that specify four integers that need to be combined using basic arithmetic operations $(+, -, x, /)$ to equal 24. For each question, we generate 6400 responses using vanilla settings and set the maximum response length per generation to 512 tokens. We use the version available at `https://huggingface.co/datasets/nlile/24-game`.

- AIME 2025. This dataset contains the 30 problems taken directly from the 2025 edition of the American Invitational Mathematics Examination (AIME). For each question, we generate 8192 responses using vanilla settings and set the maximum response length per generation to 8192 tokens.

**Experiment resources.** We used vLLM (Kwon et al., 2023) on $1 - 2$ (depending on the size of the model) NVIDIA A100 40GB GPUs per model-task pair to generate the data pools for all questions in the dataset.

**Estimating samples-to-correct.** For random sampling, we estimate the average number of samples to take (without replacement) from the data pool to find the first correct one as:

$$\text{samples-to-correct} = \frac{N + 1}{c + 1},$$

where $N$ indicates the size of the data pool and $c$ indicates the number of correct samples in the pool. Please refer to Rohatgi & Saleh (2015) for a proof. For the representation-based exploration algorithm described in Algorithm 1, we perform 5 trials per question where we record the samples-to-correct for each and take their average.

**Estimating pass@k.** To compute the pass@k values plotted in Figure 6, we follow Chen et al. (2021) and use the following unbiased estimator:

$$\text{pass@k} = \mathbb{E}_{\mathcal{D}} \left[ 1 - \frac{\binom{n-c}{k}}{\binom{n}{k}} \right] = \frac{1}{|\mathcal{D}|} \sum_{i=1}^{|\mathcal{D}|} \left[ 1 - \frac{\binom{n-c}{k}}{\binom{n}{k}} \right],$$

where $\mathcal{D}$ indicates the dataset and $|\mathcal{D}|$ indicates its size.

## B.2 DETAILS FROM SECTION 4.2

**Hyperparameters.** Similarly to Section 4.1, we initialize $\Sigma_{(0)}^{-1} = \lambda^{-1} I_d$. We set $\lambda = 0.1$. For all values of $\beta$, we use top-p $= 0.95$ and top-k $= 128$. At every time step $t$, we use a batch size of 64 to compute $h_\theta(x, y_{<t}, v_j)$ for all $v_j \in V$ where $v_j > -\infty$ (note that a logit $v_j$ is set to $-\infty$ if it gets filtered out by either the top-p or top-k filters mentioned earlier).

**Computational expense.** Note that computing the bonus $\mathbf{b}(x, y_{<t}, V)$ requires one forward pass through the model per token in the vocabulary at every time step $t$. While these can be batched together since all tokens share the same prefix, this is still prohibitively time and memory intensive. To mitigate this, we combine this method with nucleus and top-k sampling such that the bonus will only need to be computed for at most $k \ll V$ tokens.

**Dataset construction.** Due to the large computational cost of experiments, we only focus on MATH using Qwen-2.5-7b-Instruct. In addition, we do not evaluate on the full test split of MATH, but instead use a subset consisting of the 200 hardest questions as ranked by GPT-4o mini. Specifically, we sampled 1024 responses for each question in the MATH test split using GPT-4o mini to estimate the per-question pass@1. We threw out all questions for which the pass@1 was 0 (indicating not a single response was correct among all 1024), sorted the remaining questions, and kept the 200 questions with the lowest pass@1.

**Estimating samples-to-correct.**   We collected up to $1024$ generations per question (stopping early once the correct answer was found). Since the resulting samples-to-correct value found can have high variance due to the inherent randomness in the generation process, we repeat this process $5$ times with a different seed every time. This then results in a total of $200 \times 5 = 1000$ data points, minus a few data points that didn't finish running in time, for an effective total of $985$ data points used in Figure 7.

**Experiment resources.**   Experiments were run on a combination of NVIDIA A100 40GB, NVIDIA A100 80GB, and NVIDIA H100 80GB GPUs. Every run indicates one seed for a fixed question and performs up to $1024$ generations. We used one GPU (one of the several mentioned earlier) per run, adjusting the forward batch size to compute the elliptical bonus down from $64$ to $32$ when using the 40GB GPUs.

**Computing the inverse covariance.**   The inverse covariance matrix $\Sigma_{(i)}^{-1}$ includes all *mean-centered* hidden representations for all generated tokens in all $i - 1$ complete sequences generated so far for a fixed prompt $x$. Note that the mean $\mu^{(i)}$ of the raw hidden representations $h_\theta$ is computed as:

$$\mu^{(i)} = \frac{1}{H} \sum_{j=0}^{i-1} \sum_{t=1}^{T_j} h_\theta(x, y_{<t}^{(j)}), \quad H = \sum_{j=0}^{i-1} T_j,$$

where $T_j$ indicates the length (in number of tokens) of response $j$. Because of this, the mean changes after every generation $i$. This means some care is required when computing the inverse covariance matrix $\Sigma_{(i)}^{-1}$ with mean-centered hidden representations. To account for this, we will separately keep track of the inverse covariance matrix $\tilde{\Sigma}_{(i)}^{-1}$ with *non*-mean-centered hidden representations as well as the mean $\mu^{(i)}$. Then, we compute the mean-centered inverse covariance matrix $\Sigma_{(i)}^{-1}$ as

$$\Sigma_{(i)}^{-1} = \tilde{\Sigma}_{(i)}^{-1} - \left( \frac{\tilde{\Sigma}_{(i)}^{-1} \mu^{(i)} \mu_{(i)}^T \tilde{\Sigma}_{(i)}^{-1}}{-1/H + \mu_{(i)}^T \tilde{\Sigma}_{(i)}^{-1} \mu^{(i)}} \right).$$

This result is immediate from Proposition B.1.

**Proposition B.1.** *Given the current generation step $i$, the non-mean-centered inverse data covariance matrix $\tilde{\Sigma}_{(i)}^{-1}$ and the current mean $\mu^{(i)}$, the mean-centered inverse data covariance matrix $\Sigma_{(i)}^{-1}$ is given as*

$$\Sigma_{(i)}^{-1} = \tilde{\Sigma}_{(i)}^{-1} - \left( \frac{\tilde{\Sigma}_{(i)}^{-1} \mu^{(i)} \mu_{(i)}^T \tilde{\Sigma}_{(i)}^{-1}}{-1/H + \mu_{(i)}^T \tilde{\Sigma}_{(i)}^{-1} \mu^{(i)}} \right), \quad H = \sum_{j=0}^{i-1} T_j.$$

*Here, $T_j$ indicates the length (in number of tokens) of response $j$.*

**Proof of Proposition B.1.** We can write

$$\Sigma^{(i)} = \sum_{j=0}^{i-1} \sum_{t=1}^{T_j} (h_t^{(j)} - \mu_t^{(i)})(h_t^{(j)} - \mu_t^{(i)})^T$$

$$= \sum_{j=0}^{i-1} \sum_{t=1}^{T_j} h_t^{(j)}(h_t^{(j)})^T - \sum_{j=0}^{i-1} \sum_{t=1}^{T_j} h_t^{(j)}(\mu^{(i)})^T - \sum_{j=0}^{i-1} \sum_{t=1}^{T_j} \mu^{(i)}(h_t^{(j)})^T + \sum_{j=0}^{i-1} \sum_{t=1}^{T_j} \mu^{(i)}(\mu^{(i)})^T$$

$$= \tilde{\Sigma}^{(i)} - \left( \sum_{j=0}^{i-1} \sum_{t=1}^{T_j} h_t^{(j)} \right) (\mu^{(i)})^T - \mu^{(i)} \left( \sum_{j=0}^{i-1} \sum_{t=1}^{T_j} (h_t^{(j)})^T \right) + \sum_{j=0}^{i-1} \sum_{t=1}^{T_j} \mu^{(i)}(\mu^{(i)})^T$$

$$= \tilde{\Sigma}^{(i)} - H\mu^{(i)}(\mu^{(i)})^T - \mu^{(i)} H(\mu^{(i)})^T + H\mu^{(i)}(\mu^{(i)})^T$$

$$= \tilde{\Sigma}^{(i)} - 2H\mu^{(i)}(\mu^{(i)})^T + H\mu^{(i)}(\mu^{(i)})^T$$

$$= \tilde{\Sigma}^{(i)} - H\mu^{(i)}(\mu^{(i)})^T.$$

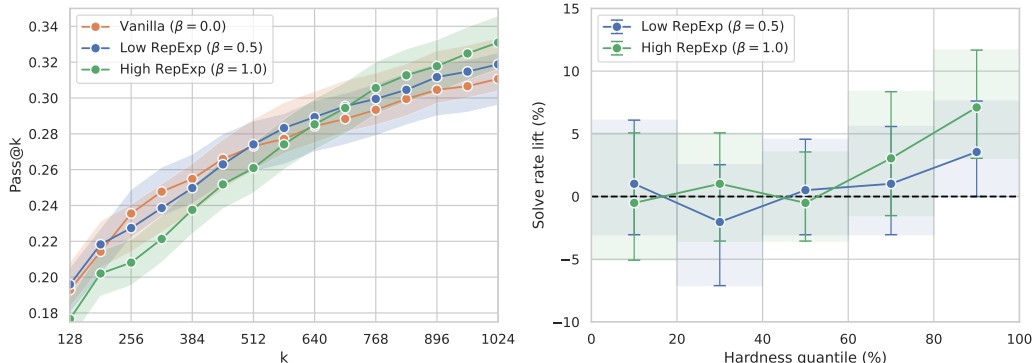

Figure 9: **Representation-based exploration at the token level**, compared to naive autoregressive generation ($\beta = 0$) for inference-time exploration. *(Left)* We added shaded areas indicating one standard error to Figure 7. *(Right)* When binning the questions by hardness (judged by samples-to-correct for GPT-4o-mini), solve rate improves the most on the hardest questions. Error bars indicate 95% paired bootstrap CIs.

Inverting $\Sigma^{(i)}$, we conclude that

$$\Sigma^{-1}_{(i)} = \left( \tilde{\Sigma}^{(i)} - H \mu^{(i)} \mu^T_{(i)} \right)^{-1}$$

$$= \tilde{\Sigma}^{-1}_{(i)} - \left( \frac{\tilde{\Sigma}^{-1}_{(i)} \mu_{(i)} \mu^T_{(i)} \tilde{\Sigma}^{-1}_{(i)}}{-1/H + \mu^T_{(i)} \tilde{\Sigma}^{-1}_{(i)} \mu^{(i)}} \right),$$

where we used the Woodbury matrix identity lemma[7] in the last step with $U = \mu^{(i)}$, $C = -H$, and $V = \mu^T_{(i)}$. □

We exclude representations from the current sequence $i$ in $\Sigma^{(i)}$ to keep the generation from veering off topic, and update $\Sigma^{-1}_{(i)}$ after each generation $i$ using $T_i$ consecutive applications of the Sherman-Morrison update in Algorithm 1, one for each hidden representation of the output tokens. Finally, we found it necessary for numerical stability to perform all covariance related computations in double precision.

**Additional plots.**   In Figure 9, we provide a revised version of Figure 7 where we add shaded areas indicating one standard error to the left plot, and we additionally add $\beta = 0.5$ to the right plot.

## C   DETAILS FOR RL POST-TRAINING EXPERIMENTS (SECTION 5)

**Hyperparameters.**   We use verl for training (Sheng et al., 2024), and provide a full overview of all common hyperparameters in Table 1, all hyperparameters specific to unlikeliness in Table 2, and all hyperpameters specific to RepExp in Table 3. Note that for AIME 2024, we adjusted the maximum prompt length to 2048, the maximum response length to 8192, the train batch size to 512, the ppo mini batch size to 128, and the ppo micro batch size per gpu to 8. Also, for RepExp on AIME 2024, we increased the sparse projection dimension from 32 to 128.

**Algorithm details.**   We note that we mean-center the representations $\bar{h}_\theta$ that are used to compute the elliptic bonuses as described in Section 3, where the mean is taken over all the response-level representations of the current group of rollouts for a fixed prompt $x$. In addition, we do not add a bonus for questions where *all* rollouts in the batch are incorrect, as we found this to empirically hurt performance.

**Dataset splits.**   For MATH, we use the original 7.5k train split for training, MATH-500 (Lightman et al., 2023) for validation, and the 4.5k (originally 5k minus the problems in MATH-500)) test split

---

[7]https://en.wikipedia.org/wiki/Woodbury_matrix_identity

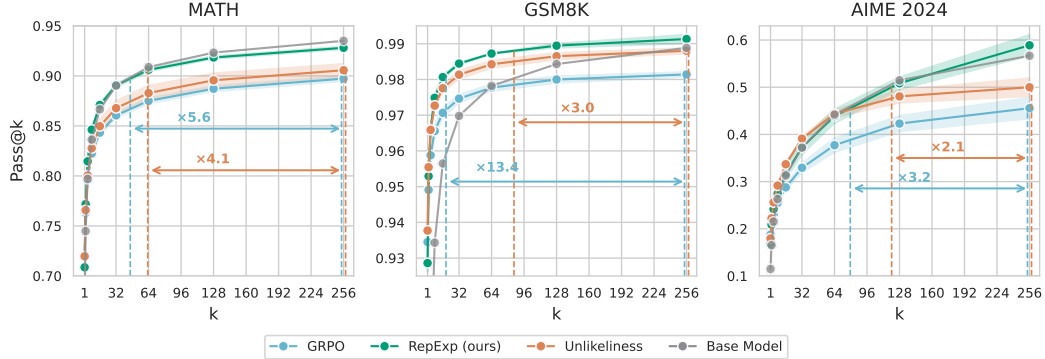

Figure 10: **Pass@k for RL post-training with exploration.** We plot Figure 2 on linear axes to provide an additional perspective.

Table 1: **Common RL hyperparameters.**

| Hyperparameter | Value |
|---|---|
| Learning rate | 1e−6 |
| Maximum prompt length | 1024 |
| Maximum response length (tokens) | 1024 |
| Train batch size | 1024 |
| PPO mini batch size | 256 |
| PPO micro batch size per gpu | 16 |
| PPO epochs | 1 |
| KL loss coefficient | 0.0 |
| GRPO group size (rollouts) | 8 |
| Entropy coefficient | 0.0 |
| Log prob micro batch size per gpu | 16 |
| Tensor model parallel size | 2 |
| Number of GPUs | 8 |
| Validation frequency | 20 |

Table 2: **Unlikeliness hyperparameters.**

| Hyperparameter | Value |
|---|---|
| $\beta_{\text{rank}}$ | 0.25 |
| No bonus if all rollouts correct | True |

Table 3: **RepExp hyperparameters.**

| Hyperparameter | Score |
|---|---|
| $\beta$ | 0.01 |
| Sparse projection dimension | 32 |
| No bonus if all rollouts incorrect | True |

for testing. For GSM8K, we use the original 8.79k train split for training, the first 512 examples from the test split for validation, and the remaining 807 examples from the test split for testing. Finally for AIME 2024, we use 4096 examples randomly chosen from the full DAPO-Math-17K dataset for training and use the full AIME 2024 dataset both for validation and testing.

**Modifications to unlikeliness baseline.** The original unlikeliness method from He et al. (2025) combines the reward modification described in Section 5 along with several other modifications to the underlying GRPO mechanics:

- They only include samples for which the resulting rollouts have nonzero advantages in the batch sent for training. Specifically, questions where either none of the rollouts are correct or all of the rollouts are correct are thrown out. To ensure batch sizes stay roughly equal, the authors implement a buffer mechanism that collects samples until the buffer reaches a target batch size.

- They increase the number of ppo epochs from 1 to 2 as they find this helps further increase the pass@k.

- They use a high KL penalty coefficient of 0.1 as they find this helps prevent the pass@k from decreasing.

Since our primary aim in this section is to isolate and compare the exploration mechanisms of our method with others, we leave out the additional changes described above when running the unlikeliness baseline. Furthermore, this allows us to use the exact same underlying GRPO hyperparameters for all methods, making the comparison much more clean.

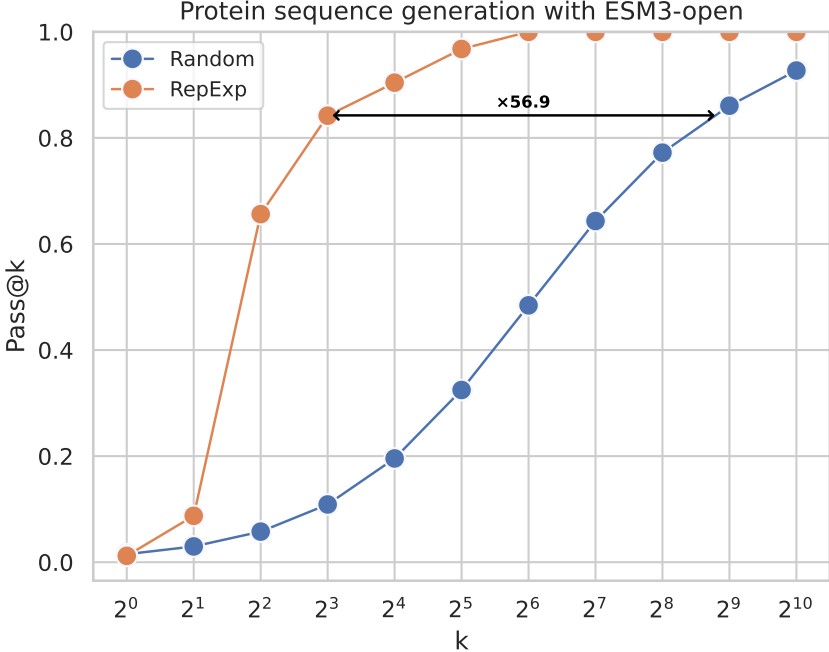

Figure 11: **Benefits of `RepExp` for protein sequence generation.** We plot the pass@$k$ curve for random vs. `RepExp` when performing inference-time selection on protein sequences from `ESM3-open`. **Compared to random sampling, `RepExp` provides up to 56.9x improvements in verifier efficiency.**

**Checkpoint picking.** For each seed per method, we pick the checkpoint during training that achieves the highest pass@1 on the respective task's validation set and use the resulting checkpoint for evaluation.

**Evaluation.** We evaluate each final checkpoint (picked in the way described earlier) on the test split of each respective task by sampling 256 responses per question using vanilla sampling parameters ($\tau = 1.0$, top-p $= 1.0$). We then estimate the pass@k exactly as described in Appendix B.1. We run 3 seeds for all methods on all tasks and average the resulting pass@k curves. In addition, we provide an alternative view of Figure 2 in Figure 10.

**Experiment resources.** We run all methods on all tasks using 8 NVIDIA H100 80GB GPUs per seed per method for 1 day.

**Computational overhead.** On `MATH`, we estimated `RepExp` to reach about $0.73$x the steps/hour throughput of GRPO. In other words, `RepExp` requires about $1.37$x higher wall-clock time per step. Note that this overhead comes from an extra forward pass that's required to compute the hidden representations of the sampled responses at each iteration of the algorithm. However, if either (1) one were to use a KL constraint (which we do not in our experiments) or (2) one were to use the hidden representations from $\pi_t$ (the current policy iterate) instead of $\pi_{\text{ref}}$ (the base model), then the extra forward pass would not be needed and `RepExp` would have the same throughput as GRPO.

## D  PROTEIN SEQUENCE GENERATION

As pointed out in Remark 2.1, we believe inference-time exploration can be useful in its own right for domains where verification is expensive. While an extensive study of this merits its own paper, we provide some preliminary experiments in the domain of protein sequence generation. In this domain, verification requires lab work, which is much more time consuming than sampling protein sequences from a generative protein model. Since we can't actually perform the lab work, we use two proxy metrics as described below.

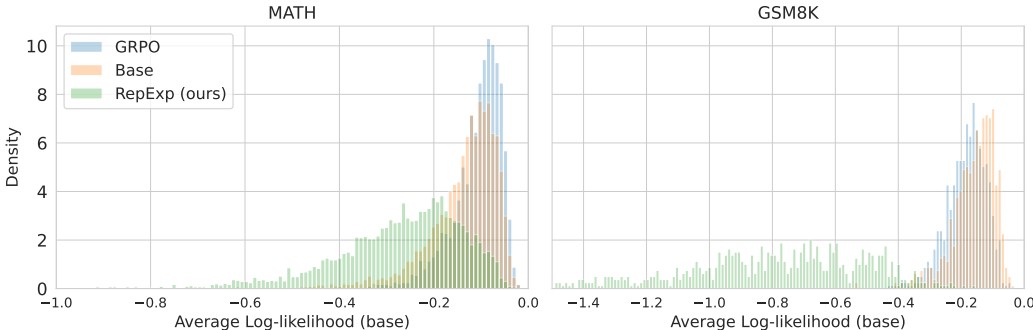

Figure 12: **Anti-sharpening behavior of RepExp.** We plot the histogram of average log-likelihoods evaluated under the base model for test responses sampled from base, GRPO, and RepExp. While the response likelihoods from GRPO either roughly stay the same or increase, the responses from RepExp are significantly more novel as indicated by the low log-likelihoods under base.

For our experimental setup, we follow the unconditional generation setup in Hayes et al. (2025) and sampled 2048 protein sequences for fully masked sequence prompts ranging in length from 64 to 916 with increments of size 4 (i.e. 64, 68, ..., 916) using ESM3-open. Then, we perform structure prediction for all sampled sequences using ESMFold (Lin et al., 2023), which returns pLDDT and pTM scores per sequence. We count a sampled sequence as plausible when pLDDT > 0.8 and pTM > 0.8. To get representations for every sequence, we use esmc-600m, the 600M parameter version of ESM C (Hayes et al., 2025; ESM Team, 2024), and average the last layer hidden representations. Finally, we perform inference-time selection for every prompt and its 2048 corresponding sampled sequences, using both random selection and RepExp. We find the average samples-to-correct for random to be 240.3 and for RepExp to be 6.7. This corresponds to a 35.9x verifier efficiency improvement. We also plot the corresponding pass@k curves in Figure 11, for which we find up to 56.9x improvements in verifier efficiency.

We find these results to be very promising and suggestive that our method can be practical in a domain where verification is genuinely expensive.

## E   BEYOND SHARPENING: FULL RESULTS

In Figure 12, we provide results for both MATH and GSM8K. We find that on GSM8K the shift toward lower likelihood responses from RepExp as evaluated under the base model is even more dramatic.

