# OpenReview forum: "Representation-Based Exploration for Language Models:  From Test-Time to Post-Training"
_ICLR.cc/2026/Conference — ICLR 2026 Poster_

### Official Review · Reviewer_iReu · 2025-10-30

**Soundness:** 3
**Presentation:** 4
**Contribution:** 4
**Rating:** 8
**Confidence:** 4

**Summary:**

This paper focuses on the problem of exploration in the context of LLM reasoning. Existing RL methods are known to “sharpen” the distribution of responses v/s learning novel strategies for reasoning. To overcome this limitation, the authors propose RepExp—a simple yet principled exploration technique that leverages the internal representations of LLMs. Inspired by the literature of exploration in deep RL, RepExp encourages diversity in reasoning by providing an elliptical exploration bonus based on the model’s internal representations. The authors demonstrate the effectiveness of RepExp in two distinct settings: (1) a novel inference-time selection task, and (2) RL-based post-training. Across multiple datasets and problem types, RepExp consistently outperforms existing methods.

**Strengths:**

1. The paper is clearly written and highly accessible. Moreover, the exploration problem in the context of LLM reasoning is both timely and highly relevant, addressing a critical challenge within LLM reasoning.

2. The paper’s introduction of the inference-time selection task—serving both as an evaluation tool for exploration and as a standalone research problem—is an interesting and valuable addition to the study.

3. The proposed method RepExp is well motivated.

4. The results presented in this paper are both strong and compelling, and are effectively communicated through the well-structured Research Findings (RFs).

**Weaknesses:**

1. Section 2.1: (Line 143) ““Maximally diverse and high-probability of containing a positive response”: The inference-time selection problem is defined as selecting generations based on both diversity and quality. However, from my understanding, RepExp appears to select only for diversity. Could this lead to cases where the LLM produces hallucinated or incorrect reasoning that is nonetheless selected by RepExp due to its diversity?

2. In the RL post-training setting, the covariance matrix is computed using the hidden reprs of all the responses first, and then used to calculate div(x,y). From the motivation of elliptical bonuses (Lines 201-204), div(h,h<i), is the prediction error for a new “h” which is not in the training set. In this setting, isn’t “h” already in the training set? Is this approach then still principled? (prediction error is bounded by div(h|h<i))

3. “this method is history aware” – this line is a bit unclear, as it could be interpreted as div(x,y) is also conditioned on past-optimisation timesteps (which I assume is not the case). I think “history” can be a bit confusing for referring to responses which are often generated in parallel (which are therefore assigned the same generation timestep).

4. How do other exploration-based methods, such as unlikeness or entropy, perform on the inference-time selection tasks? As the authors note, this setting provides a nice way to isolate and assess the impact of the exploration bonus.

5. (Line 403) “not optimized … hence does not yet give a wall-clock time improvement“ – doesn’t this method have a complexity of k (forward passes) * k generations * T? How can this be improved to be faster than naive autoregressive generation?

**Questions:**

1. Have you considered persisting the covariance matrix across training generations? — Can the LLM then explore novel strategies that are applicable across multiple problems?

2. Does RepExp lead to “novel” reasoning patterns? Moreover, is there a way to quantitatively assess the diversity of the generated responses—for instance, analogous to how state coverage is used in reinforcement learning?

---

> ### Author Response · Authors · 2025-11-23
> **Response to Reviewer iReu**
>
> We thank the reviewer for the valuable feedback.
>
> > Section 2.1: (Line 143) ““Maximally diverse and high-probability of containing a positive response”: The inference-time selection problem is defined as selecting generations based on both diversity and quality. However, from my understanding, RepExp appears to select only for diversity. Could this lead to cases where the LLM produces hallucinated or incorrect reasoning that is nonetheless selected by RepExp due to its diversity?
>
> This is a good point, and RepExp indeed sometimes latches on to highly diverse but nevertheless nonsensical generations. However, we’re mostly leaning on the LLM itself as providing some baseline level of quality in its responses. This also explains why:
> - RepExp doesn’t work well for weaker models (figure 3, left)
> - RepExp doesn’t work well on top of a data pool that was generated with high-temperature sampling (figure 4, purple), as high-temperature sampling tends to induce technically more diverse, but often incorrect, degenerate, or non-parsable responses.
>
> > In the RL post-training setting, the covariance matrix is computed using the hidden reprs of all the responses first, and then used to calculate div(x,y). From the motivation of elliptical bonuses (Lines 201-204), div(h,h<i), is the prediction error for a new “h” which is not in the training set. In this setting, isn’t “h” already in the training set? Is this approach then still principled? (prediction error is bounded by div(h|h<i))
>
> Yes, this is still principled–let div(h,{h<i, h}) denote the value of div(h, h_{h<i}) when $h$ is included in the training set. For the specific elliptic bonus we use, a standard argument in the bandit literature shows that when $\lambda \geq 1$ and $||h|| \leq 1$,
>
> $c * div(h, h_{h<i}) \leq div(h,{h<i, h}) \leq C * div(h, h_{h<i})$ for absolute constants $c$ and $C$.
>
> In other words, up to constants,  div(h,{h<i, h}) is equivalent to a “clipped” version of the more standard bonus, which suffices to achieve the same theoretical guarantees.
>
> **See Lemma 1 here**: https://arxiv.org/pdf/2010.10182
>
> The quantity div(h,{h<i, h}) also has an appealing interpretation as a “leverage score” in statistics, see https://en.wikipedia.org/wiki/Leverage_(statistics). Note that leverage scores enjoy nice properties such as being bounded between 0 and 1, which makes our bonus better behaved and allows us to add it to the extrinsic reward without any additional normalization tricks. We will make this more clear in the camera-ready version.
>
> > “this method is history aware” – this line is a bit unclear, as it could be interpreted as div(x,y) is also conditioned on past-optimisation timesteps (which I assume is not the case). I think “history” can be a bit confusing for referring to responses which are often generated in parallel (which are therefore assigned the same generation timestep).
>
> We understand the confusion here - the “history-aware” term is referring to RepExp in the inference-time setting, where responses are selected serially and the covariance matrix is updated incrementally with the representations of all the past responses that have already been selected. In the RL setting, “group-aware” would be more descriptive since in this setting the generations are indeed done in parallel. We’ll be sure to update the wording in the camera-ready version of the paper in light of this discussion.
>
> > How do other exploration-based methods, such as unlikeness or entropy, perform on the inference-time selection tasks? As the authors note, this setting provides a nice way to isolate and assess the impact of the exploration bonus.
>
> We considered including some inference-time versions of these, but  decided against it, as it would require arguably unfaithful manipulations to these methods. For Unlikeliness, which inherently depends on an extrinsic reward in its published form, the best inference-time equivalent involves serially picking the least likely response still left in the pool, which almost certainly results in poor performance. Similarly, there are many ways to incorporate entropy into inference-time selection, but none of them seem like they would do very well and hence we didn’t think it would be worth including them. That said, we'd be happy to add some of these if the reviewer has specific inference-time versions in mind that would be valuable to include.

---

> ### Author Response · Authors · 2025-11-23
> **Response to Reviewer iReu (continued)**
>
> > (Line 403) “not optimized … hence does not yet give a wall-clock time improvement“ – doesn’t this method have a complexity of k (forward passes) * k generations * T? How can this be improved to be faster than naive autoregressive generation?
>
> We apologize for the confusion here, let us try to clarify: For a fixed, large k, RepExp obtains better pass@k performance than naive autoregressive generation. We might expect that this directly translates to a wall-clock time improvement, as RepExp can achieve the same pass@k performance as naive autoregressive generation with fewer attempts. However, since our implementation of RepExp requires an extra forward pass per generated token to compute the hidden states of candidate next tokens to be generated, there isn’t actually any wall-clock time improvement in practice. Our results are provided as a proof of concept, and we believe alleviating this requirement - obtaining reasonable representations without additional forward passes - to be an important area for future research.
>
> Indeed, if additional forward passes were not necessary, we would expect a wall-clock time improvement as well (due to the sample-efficiency improvement of RepExp, not due to an inherently faster implementation than naive autoregressive generation). We’ll be sure to update the wording in the camera-ready version of the paper to reflect this clarification.
>
> > Have you considered persisting the covariance matrix across training generations? — Can the LLM then explore novel strategies that are applicable across multiple problems?
>
> Yes, we considered having one covariance matrix per question in the dataset and then persisting this covariance whenever the same question is sampled. However, we found the performance to degrade quite quickly, and hence we did not pursue this further.
>
> > Does RepExp lead to “novel” reasoning patterns? Moreover, is there a way to quantitatively assess the diversity of the generated responses—for instance, analogous to how state coverage is used in reinforcement learning?
>
> We refer the reviewer to part 1 of the general response, for which we restate the most relevant part below:
>
> We agree there could be more analysis to see if the model post-trained with RepExp produces behaviors beyond sharpening. To this end, inspired by figure 4 of [1], we have included some additional results where we sample a single response from the base model, the GRPO post-trained model, and the RepExp post-trained model on the full test sets of MATH and GSM8K. We then score all responses under the base model in terms of log likelihood. We plot the resulting histograms [here (for MATH)](https://anonymous.4open.science/r/rep-exp-5594/histogram_math_1.0_hard_1.0_seed_41.pdf) and [here (for GSM8k)](https://anonymous.4open.science/r/rep-exp-5594/histogram_gsm8k_1.0_hard_1.0_seed_41.pdf). We find that in both domains the responses from RepExp tend to be less likely under the base model, quite dramatically so on GSM8K, indicating it’s generating responses that look more novel under the base model and hence not merely performing sharpening. In contrast, notice that standard GRPO exhibits sharpening behavior on MATH, as demonstrated by the movement of probability mass towards the right with respect to the base model.
>
> [1] Karan, Aayush, and Yilun Du. "Reasoning with sampling: Your base model is smarter than you think." arXiv preprint arXiv:2510.14901 (2025).
>
> We hope our rebuttal clarifies the reviewer’s confusions and relieves some of their concerns.

---

> > ### Comment · Reviewer_iReu · 2025-11-26
> >
> > Thank you for the comprehensive responses. All of my concerns have now been resolved.

---

> > > ### Author Response · Authors · 2025-11-26
> > >
> > > We're glad to hear we were able to address all the reviewer's concerns. Thank you for your time and active engagement in the rebuttal process.

---

### Official Review · Reviewer_VhJg · 2025-10-30

**Soundness:** 2
**Presentation:** 2
**Contribution:** 2
**Rating:** 2
**Confidence:** 4

**Summary:**

•	Principled and simple approach: The use of elliptical bonuses derived from model representations is conceptually clean and grounded in existing exploration theory, avoiding the complexity of auxiliary networks.
•	Broad empirical coverage: The paper evaluates across multiple tasks, datasets, and model families, providing a comprehensive empirical picture.
•	Clear dataset justification: The chosen benchmarks are well-motivated and represent a diverse set of reasoning and coding tasks.

**Strengths:**

•	Principled and simple approach: The use of elliptical bonuses derived from model representations is conceptually clean and grounded in existing exploration theory, avoiding the complexity of auxiliary networks.
•	Broad empirical coverage: The paper evaluates across multiple tasks, datasets, and model families, providing a comprehensive empirical picture.
•	Clear dataset justification: The chosen benchmarks are well-motivated and represent a diverse set of reasoning and coding tasks.

**Weaknesses:**

•	Overstated claims relative to results: The introduction makes strong assertions about “moving beyond sharpening” and “substantial improvements,” but empirical gains are modest and often inconsistent—particularly in RL post-training, where RepExp sometimes performs on par or worse than baselines (e.g., Figure 6).
•	Limited discussion of negative results: The method degrades performance for weaker models, yet this is not adequately analyzed or contextualized.
•	Unclear motivation and structure: The introduction is hard to follow due to numerous forward references and misplaced motivation (appearing in Section 2). The “Contributions” subsection mixes results with claims, making it difficult to distinguish novelty from outcomes.
•	Lack of practical relevance: The benefits at inference-time (reducing verifier calls) are meaningful only when verifier cost dominates, but the paper fails to demonstrate this in a realistic application scenario.
•	Inconclusive post-training value: For large k, RepExp sometimes underperforms Unlikeliness or GRPO; thus, it is unclear when and why this exploration actually helps.
•	Insufficient alignment between claims and framing: The paper promises exploration as a path toward discovering new capabilities, but the experiments focus narrowly on pass@k efficiency, not on qualitatively novel behaviors.

**Questions:**

1.	Can the authors clarify why weaker models degrade under representation-based exploration, and whether this correlates with representational quality or another factor?
2.	In the RL setting, what is the computational overhead of computing bonuses compared to standard GRPO training?

---

> ### Author Response · Authors · 2025-11-23
> **Response to Reviewer Vhjg**
>
> We thank the reviewer for the valuable feedback.
>
> > Overstated claims relative to results: The introduction makes strong assertions about “moving beyond sharpening” and “substantial improvements,” but empirical gains are modest and often inconsistent—particularly in RL post-training, where RepExp sometimes performs on par or worse than baselines (e.g., Figure 6).
>
> We refer the reviewer to part 1 of our general response, which we restate below for convenience:
>
> Our work is meant to be **a stepping stone towards moving beyond the sharpening regime** and discovering qualitatively new behaviors. To this end, we first test whether a representation-based approach can guide the search for diverse behaviors in our novel, inference-time selection sandbox environment, after which we demonstrate that improvements in this setting extend to the RL setting as well. We believe the fact that this bonus prevents the diversity collapse phenomenon in RL is a promising signal indicating that deliberate exploration is a practical path toward the discovery of new behaviors beyond sharpening.
>
> Nevertheless, we agree there could be more analysis to see if the model post-trained with RepExp produces behaviors beyond sharpening. To this end, inspired by figure 4 of [1], we have included some additional results where we sample a single response from the base model, the GRPO post-trained model, and the RepExp post-trained model on the full test sets of MATH and GSM8K. We then score all responses under the base model in terms of log likelihood. We plot the resulting histograms [here (for MATH)](https://anonymous.4open.science/r/rep-exp-5594/histogram_math_1.0_hard_1.0_seed_41.pdf) and [here (for GSM8k)](https://anonymous.4open.science/r/rep-exp-5594/histogram_gsm8k_1.0_hard_1.0_seed_41.pdf). We find that in both domains the responses from RepExp tend to be less likely under the base model, quite dramatically so on GSM8K, indicating it’s generating responses that look more novel under the base model and hence not merely performing sharpening. In contrast, notice that standard GRPO exhibits sharpening behavior on MATH, as demonstrated by the movement of probability mass towards the right with respect to the base model.
>
> With respect to the RL post-training results, we’d like to emphasize that **for large k (2^6 - 2^8), RepExp performs better than all baselines in all tasks** (except on par for AIME 2024 and k = 2^6). In addition, for small k, we argue the degradation is rather minor since:
> - With respect to GRPO, it’s only for k = 1 that RepExp slightly underperforms. As soon as k >= 2, it’s on par or better than GRPO in all domains.
> - From a sample-efficiency perspective, it only takes one extra generation to more than make up the gap in pass@1. **However, on the large k side, it often takes hundreds of extra generations from either Unlikeliness or GRPO to match RepExp**. For example, on AIME 2024, GRPO needs 256 samples to reach the same level of performance as RepExp with only 80 samples.
>
> [1] Karan, Aayush, and Yilun Du. "Reasoning with sampling: Your base model is smarter than you think." arXiv preprint arXiv:2510.14901 (2025).
>
> > Limited discussion of negative results: The method degrades performance for weaker models, yet this is not adequately analyzed or contextualized.
>
> We would like to gently push back on the claim that the degradation of weaker models is not adequately analyzed or contextualized, as we have dedicated one of our main figures (figure 3, left) to highlight the relation between model strength and improvement due to exploration. We additionally analyze and describe these results in RF2 as part of section 4.1. That being said, if the reviewer has any specific additional analysis they would like to see, we’d be happy to add it.
>
> Furthermore, as noted in part 3 of our general response, we emphasize that **we see the degradation of exploration performance for weaker models as a finding of our work, not a weakness**. One of the main contributions of our work is better understanding when and how representation-based exploration works, and we expect the finding that model strength plays an important role here to inform future work studying representation-based exploration.

---

> > ### Author Response · Authors · 2025-11-23
> > **Response to Reviewer Vhjg (continued)**
> >
> > > Unclear motivation and structure: The introduction is hard to follow due to numerous forward references and misplaced motivation (appearing in Section 2). The “Contributions” subsection mixes results with claims, making it difficult to distinguish novelty from outcomes.
> >
> > The forward references (we are assuming the reviewer is referring to section references here) are meant to provide a roadmap for the paper. This is a standard practice to improve readability. More generally, we cared a lot about the writing, which has been appreciated by several reviewers who noted that “[t]he paper is clearly written and highly accessible” (reviewer iReu) and “[t]he paper is really nicely written and structured. There was a dedicated effort to make it easy to understand and follow the structure” (reviewer gWc2). That being said, we’re happy to take specific suggestions to minimize confusion.
> >
> > Regarding the contributions subsection, we intentionally mentioned specific results as evidence that supports stated claims. That being said, we will be adjusting some of the language in a way that we think improves clarity in the camera-ready version of the paper. Specifically, we plan to remove some of the explicit “results” language in some of the bolded headings. As mentioned earlier, we would be happy to accommodate specific requests that might improve clarity.
> >
> > > Lack of practical relevance: The benefits at inference-time (reducing verifier calls) are meaningful only when verifier cost dominates, but the paper fails to demonstrate this in a realistic application scenario.
> >
> > We refer the reviewer to part 2 of our general response, which we restate below for convenience:
> > While we agree it would be interesting to investigate our method in domains where verification is more expensive, our paper uses standard datasets to tractably verify the improvement in efficiency by analogy. We believe our experiments on various tasks and model-dataset pairs are extensive. Still, we’d love to see future work evaluate our method in settings like RLHF or protein sequence generation with language models where verification is known to be inherently expensive. As mentioned by reviewer gWc2, experiments of this sort are somewhat out of scope and probably deserve their own full paper.
> >
> > In addition, we’d also like to remind the reviewer that although we believe our inference-time version of RepExp is indeed practical in domains where verification is expensive, the primary purpose of the inference-time selection setting is as a **sandbox to test exploration methods in isolation, without interference from complex RL mechanisms such as optimization and generalization**. The intention is that this sandbox setting can help the design of future exploration methods to be applied during post-training time.
> >
> > > Inconclusive post-training value: For large k, RepExp sometimes underperforms Unlikeliness or GRPO; thus, it is unclear when and why this exploration actually helps.
> >
> > This is not correct: when k is anywhere 2^6 - 2^8, RepExp performs better than both Unlikeliness and GRPO in all tasks (except on par for AIME 2024 and k = 2^6). **Hence, our results consistently indicate an advantage of our method over GRPO and Unlikeliness for large k**.

---

> > > ### Author Response · Authors · 2025-11-23
> > > **Response to Reviewer Vhjg (continued)**
> > >
> > > > Insufficient alignment between claims and framing: The paper promises exploration as a path toward discovering new capabilities, but the experiments focus narrowly on pass@k efficiency, not on qualitatively novel behaviors.
> > >
> > > We refer the reviewer to our answer to the very first concern we addressed above (“Overstated claims relative to results …”) as well as to part 1 of our general response. We restate our answer below for convenience:
> > >
> > > Our work is meant to be **a stepping stone towards moving beyond the sharpening regime** and discovering qualitatively new behaviors. To this end, we first test whether a representation-based approach can guide the search for diverse behaviors in our novel, inference-time selection sandbox environment, after which we demonstrate that improvements in this setting extend to the RL setting as well. We believe the fact that this bonus prevents the diversity collapse phenomenon in RL is a promising signal indicating that deliberate exploration is a practical path toward the discovery of new behaviors beyond sharpening.
> > >
> > > Nevertheless, we agree there could be more analysis to see if the model post-trained with RepExp produces behaviors beyond sharpening. To this end, inspired by figure 4 of [1], we have included some additional results where we sample a single response from the base model, the GRPO post-trained model, and the RepExp post-trained model on the full test sets of MATH and GSM8K. We then score all responses under the base model in terms of log likelihood. We plot the resulting histograms [here (for MATH)](https://anonymous.4open.science/r/rep-exp-5594/histogram_math_1.0_hard_1.0_seed_41.pdf) and [here (for GSM8k)](https://anonymous.4open.science/r/rep-exp-5594/histogram_gsm8k_1.0_hard_1.0_seed_41.pdf). We find that in both domains the responses from RepExp tend to be less likely under the base model, quite dramatically so on GSM8K, indicating it’s generating responses that look more novel under the base model and hence not merely performing sharpening. In contrast, notice that standard GRPO exhibits sharpening behavior on MATH, as demonstrated by the movement of probability mass towards the right with respect to the base model.
> > >
> > > [1] Karan, Aayush, and Yilun Du. "Reasoning with sampling: Your base model is smarter than you think." arXiv preprint arXiv:2510.14901 (2025).
> > >
> > > > Can the authors clarify why weaker models degrade under representation-based exploration, and whether this correlates with representational quality or another factor?
> > >
> > > We indeed hypothesize that weaker models tend to have worse representations, which in turn decreases the RepExp performance, as it explicitly relies on similarities and differences in representations being linguistically meaningful.
> > >
> > > > In the RL setting, what is the computational overhead of computing bonuses compared to standard GRPO training?
> > >
> > > Compared to standard GRPO training, RepExp requires an extra forward pass through the reference model for every fully generated response to get the final hidden states that the bonus rewards are computed from. Practically, these forward passes are batched together for efficiency.
> > >
> > > We hope our rebuttal clarifies the reviewer’s confusions and relieves some of their concerns.

---

> > > > ### Comment · Reviewer_VhJg · 2025-11-24
> > > >
> > > > Thanks to the authors for the rebuttal. A few of my earlier points were clarified:
> > > >
> > > > - I agree that the degradation on weaker models was already explained sufficiently in Section 4.1.
> > > > - Regarding runtime, the clarification that RepExp requires an additional forward pass is helpful. I would still encourage the authors to explicitly quantify this overhead in the paper (e.g., relative wall-clock cost), so that readers can clearly assess the practical implications.
> > > >
> > > > That said, my main concern about the actual effectiveness of the method, especially in the post-training setting, remains. Even with the additional clarification, the results still look quite mixed. For instance, on MATH, unlikelihood performs similarly or better for very small \(k\), and for large \(k\) the base model achieves a higher performance. On AIME, RepExp only really outperforms the baselines for very large \(k\).

---

> ### Author Response · Authors · 2025-11-24
> **Response to Reviewer VhJG**
>
> We thank the reviewer for their active engagement in the rebuttal! We’re happy to hear we were able to address their concerns.
>
> > I would still encourage the authors to explicitly quantify this overhead in the paper (e.g., relative wall-clock cost), so that readers can clearly assess the practical implications.
>
> Thank you for this suggestion. We’ll be sure to update the camera-ready version of our paper with concrete wall-clock numbers.
>
> > That said, my main concern about the actual effectiveness of the method, especially in the post-training setting, remains. Even with the additional clarification, the results still look quite mixed. For instance, on MATH, unlikelihood performs similarly or better for very small (k), and for large (k) the base model achieves a higher performance. On AIME, RepExp only really outperforms the baselines for very large (k).
>
> *“..., on MATH, unlikelihood performs similarly or better for very small (k)”*
>
> We emphasize that the x-axis here is on a log scale. Hence, the reviewer’s claim is only true for k = 1 (where RepExp *slightly* underperforms Unlikeliness) and k = 2 (where RepExp performs as well as  Unlikeliness). **For *any* k > 2, RepExp strictly outperforms both Unlikeliness and GRPO on MATH**. We have linked [versions of our plots with linear axes here](https://anonymous.4open.science/r/rep-exp-5594/rl_pass_at_k_best_pass@1_linear_axes.pdf) to better highlight these differences.
>
> *“..., and for large (k) the base model achieves a higher performance”*
>
> While we think the comparison between our method and the base model for large k in MATH is better described as “roughly on par”, we understand the reviewer’s concern. However, note that **both GRPO and Unlikelihood severely underperform the base model for large k in *all* domains, while RepExp does not suffer from this “diversity collapse” [1] phenomenon in *any* of the domains**.
>
> *“On AIME, RepExp only really outperforms the baselines for very large (k).”*
>
> First, we’d like to point out that the large k regime is most important—from a sample-efficiency perspective, it often only takes one or two generations to more than make up the gap in pass@k when k is very small. **However, on the large k side, it often takes hundreds of extra generations from either Unlikeliness or GRPO to match RepExp**. For example, on AIME 2024, GRPO needs 256 samples to reach the same level of performance as RepExp with only 80 samples.
>
> Second, we emphasize again that the x-axis here is on a log scale. So while it might seem that there are more points where RepExp underperforms Unlikeliness, we note that there are many more points from 2^6 - 2^8 than from 2^0 - 2^6. To make this point clear, we refer the reviewer to [this plot where we use a linear scale on the x-axis](https://anonymous.4open.science/r/rep-exp-5594/rl_pass_at_k_best_pass@1_linear_axes.pdf) for all tasks.
>
> **To summarize, the consistent advantage of RepExp *across* domains is that it is competitive with both GRPO and Unlikeliness at small k, while at the same time outperforming both at large k**. This is because both GRPO and Unlikeliness suffer from the “diversity collapse” phenomenon [1] (i.e. degrade at large k), while RepExp does not. While RepExp is on par with the base model at large k, it crucially achieves this while still enjoying comparatively big lifts at small k.
>
> Finally, we encourage the reviewer to also see these results in light of the broader intentions of this paper. Our goal is not to chase state-of-the-art numbers on the benchmarks we use, but to investigate how we might drive active exploration in LLMs, starting with a simple but well-established method and a carefully designed inference-selection task. While we didn’t find our method to be necessarily uniformly better than all baselines at all tasks in the RL post-training setting, our results still show extremely strong positive signal both at small k (where RepExp performs close to competitive baselines) and at large k (where RepExp outperforms all other RL baselines by preventing diversity collapse). We believe these results are  encouraging and hope they inspire future work in this area.
>
> We hope the clarifications above make the value and contributions of our RL post-training results more clear. We would greatly appreciate it if the reviewer could reconsider their score if this is their only remaining concern.
>
> [1] Yue, Yang, et al. "Does reinforcement learning really incentivize reasoning capacity in llms beyond the base model?." arXiv preprint arXiv:2504.13837 (2025).

---

> > ### Comment · Reviewer_VhJg · 2025-11-25
> >
> > I thank the authors for the additional effort and the helpful clarifications. As most of my concerns have now been addressed, I am happy to raise my score to 4.

---

> ### Author Response · Authors · 2025-11-25
>
> We thank the reviewer again for their active involvement in the rebuttal process --- we really appreciate it.
>
> While we’re happy to see the reviewer has increased their score, we’d like to follow up and ask the reviewer to consider increasing their score further to above the acceptance threshold given that their concerns have been addressed.
>
> Thank you again for your time!

---

> > ### Author Response · Authors · 2025-12-01
> > **Additional Results with Expensive Verification (Protein Sequence Generation)**
> >
> > We wanted to follow up with the reviewer regarding the following concern:
> >
> > > Lack of practical relevance: The benefits at inference-time (reducing verifier calls) are meaningful only when verifier cost dominates, but the paper fails to demonstrate this in a realistic application scenario.
> >
> > In addition to our general response part 2, we refer the reviewer to our "Additional Results on Protein Sequence Generation" post, which we restate below for convenience:
> >
> > We have conducted some preliminary experiments in the domain of protein sequence generation, based on the suggestion from reviewer gWc2. In this domain, verification requires lab work, which is much more time consuming than sampling protein sequences from a generative protein model. Since we can’t actually perform the lab work, we use two proxy metrics as described below.
> >
> > For our experimental setup, we followed the unconditional generation setup in the ESM3 paper [1] and sampled 2048 protein sequences for fully masked sequence prompts ranging in length from 64 to 916 with increments of size 4 (i.e. {64, 68, ..., 916}). Then, we performed structure prediction for all sampled sequences using ESMFold [2], which returns pLDDT and pTM scores per sequence. We count a sampled sequence as plausible when pLDDT > 0.8 and pTM > 0.8. To get representations for every sequence, we use ESM-C [1] and average the last layer hidden representations. Finally, we perform inference-time selection for every prompt and its 2048 corresponding sampled sequences, using both random selection and RepExp. We find the average samples-to-correct for random to be 240.3 and for RepExp to be 6.7. **This corresponds to a 35.9x verifier efficiency improvement**. We also plot the [corresponding pass@k curves here](https://anonymous.4open.science/r/rep-exp-5594/protein_pass_at_k.pdf), for which **we find up to 56.9x improvements in verifier efficiency**.
> >
> > We find these results to be very promising and hope they demonstrate to the reviewer that our method can be practical in a domain where verification is genuinely expensive.
> >
> > **References**
> >
> > [1] Thomas Hayes et al., Simulating 500 million years of evolution with a language model. Science 387, 850-858 (2025).
> >
> > [2] Zeming Lin et al., Evolutionary-scale prediction of atomic-level protein structure with a language model. Science 379, 1123-1130 (2023).

---

### Official Review · Reviewer_gWc2 · 2025-10-31

**Soundness:** 3
**Presentation:** 4
**Contribution:** 4
**Rating:** 8
**Confidence:** 4

**Summary:**

The paper studies representation based exploration for language models. An timely and highly important problem. Discovery of new and diverse behaviours would be a critical component for future revolutions in many AI4 science applications for example.

The paper is exploring this question using many established based models and a decent selection of downstream tasks including AIME.

**Strengths:**

The paper is really nicely written and structured. There was a dedicated effort to make it easy to understand and follow the structure.

The models studied include Llama, Mistral, Qwen and Phi and the tasks are representative.

**Weaknesses:**

Given the importance of the question in many domains I would have loved to see a similar study for protein language models and or how far the gap is in that area. That probably will however merit its own paper and is out of scope.

Right now it states that "In addition, we have uploaded a zip file with the complete, anonymized
code for all our experiments and plots." Will the code, the data and the experimental setup be made publicly available after acceptance?

**Questions:**

See the question for code and data in weaknesses?

---

> ### Author Response · Authors · 2025-11-23
> **Response to Reviewer gWc2**
>
> We thank the reviewer for the valuable feedback.
>
> > Given the importance of the question in many domains I would have loved to see a similar study for protein language models and or how far the gap is in that area. That probably will however merit its own paper and is out of scope.
>
> We agree our findings could be relevant for AI for Science applications and would love to see future work investigating this!
>
> > Right now it states that "In addition, we have uploaded a zip file with the complete, anonymized code for all our experiments and plots." Will the code, the data and the experimental setup be made publicly available after acceptance?
>
> Absolutely! We will make everything publicly available.

---

> ### Author Response · Authors · 2025-12-01
> **Additional Results on Protein Sequence Generation**
>
> We wanted to follow up with the reviewer regarding the following comment:
>
> > Given the importance of the question in many domains I would have loved to see a similar study for protein language models and or how far the gap is in that area. That probably will however merit its own paper and is out of scope.
>
> We refer the reviewer to our "Additional Results on Protein Sequence Generation" post, which we restate below for convenience:
>
> We have conducted some preliminary experiments in the domain of protein sequence generation, based on the suggestion from reviewer gWc2. In this domain, verification requires lab work, which is much more time consuming than sampling protein sequences from a generative protein model. Since we can’t actually perform the lab work, we use two proxy metrics as described below.
>
> For our experimental setup, we followed the unconditional generation setup in the ESM3 paper [1] and sampled 2048 protein sequences for fully masked sequence prompts ranging in length from 64 to 916 with increments of size 4 (i.e. {64, 68, ..., 916}). Then, we performed structure prediction for all sampled sequences using ESMFold [2], which returns pLDDT and pTM scores per sequence. We count a sampled sequence as plausible when pLDDT > 0.8 and pTM > 0.8. To get representations for every sequence, we use ESM-C [1] and average the last layer hidden representations. Finally, we perform inference-time selection for every prompt and its 2048 corresponding sampled sequences, using both random selection and RepExp. We find the average samples-to-correct for random to be 240.3 and for RepExp to be 6.7. **This corresponds to a 35.9x verifier efficiency improvement**. We also plot the [corresponding pass@k curves here](https://anonymous.4open.science/r/rep-exp-5594/protein_pass_at_k.pdf), for which **we find up to 56.9x improvements in verifier efficiency**.
>
> We find these results to be very promising and would love to see future work take these further.
>
> **References**
>
> [1] Thomas Hayes et al., Simulating 500 million years of evolution with a language model. Science 387, 850-858 (2025).
>
> [2] Zeming Lin et al., Evolutionary-scale prediction of atomic-level protein structure with a language model. Science 379, 1123-1130 (2023).

---

### Official Review · Reviewer_ppuG · 2025-11-01

**Soundness:** 3
**Presentation:** 3
**Contribution:** 3
**Rating:** 4
**Confidence:** 4

**Summary:**

This paper investigates representation-based exploration as a means to improve reasoning and diversity in language model behavior, both at inference time and during reinforcement learning post-training. The authors propose a simple yet principled elliptic bonus derived from a model’s hidden states to encourage exploration. Across several benchmarks (MATH, GSM8K, MBPP+, Game of 24, AIME), the approach yields significant improvements in verifier efficiency and maintains performance or improves at large Pass@k. The effect holds for larger models and, specifically, harder tasks.

**Strengths:**

- Adapting elliptical bonuses to language-model representations is well motivated, interesting, conceptually elegant, and grounded in prior exploration theory.
- Comprehensive empirical evaluation across multiple model families, scales, and sampling strategies.
- Significant improvements in sample efficiency.

**Weaknesses:**

- The proposed exploration strategy significantly degrades performance for smaller language models (e.g., Qwen-0.5B, Mistral-7B).
- The paper’s main narrative emphasizes discovering novel behaviors beyond sharpening, yet the results primarily reflect improvements in verifier sample efficiency. For large k, performance remains comparable to the base model, suggesting that the method broadens coverage rather than uncovering qualitatively new capabilities. The work would benefit from being reframed explicitly as a study in verifier-efficiency optimization rather than behavioral discovery.
- In the post-training experiments, the method consistently underperforms GRPO and Unlikeliness for small values of k.
- All benchmarks employed (GSM8K, MATH, MBPP+, AIME, Game of 24) feature cheap, automatic verifiers where verification cost is negligible relative to model inference. The paper would benefit from experiments in domains where verification is more expensive

**Questions:**

- Why did you choose Qwen-2.5-7b-Instruct for the post-training experiments?

---

> ### Author Response · Authors · 2025-11-23
> **Response to Reviewer ppuG**
>
> We thank the reviewer for the valuable feedback.
>
> > The proposed exploration strategy significantly degrades performance for smaller language models (e.g., Qwen-0.5B, Mistral-7B).
>
> We refer the reviewer to part 3 of our general response, which we restate below for convenience:
> We emphasize that **we see the degradation of exploration performance for weaker models as a finding of our work, not a weakness**. One of the main contributions of our work is better understanding when and how representation-based exploration works, and we expect the finding that model strength plays an important role here to inform future work studying representation-based exploration.
>
> > The paper’s main narrative emphasizes discovering novel behaviors beyond sharpening, yet the results primarily reflect improvements in verifier sample efficiency. For large k, performance remains comparable to the base model, suggesting that the method broadens coverage rather than uncovering qualitatively new capabilities. The work would benefit from being reframed explicitly as a study in verifier-efficiency optimization rather than behavioral discovery.
>
> We refer the reviewer to part 1 of our general response, which we restate below for convenience:
>
> Our work is meant to be **a stepping stone towards moving beyond the sharpening regime** and discovering qualitatively new behaviors. To this end, we first test whether a representation-based approach can guide the search for diverse behaviors in our novel, inference-time selection sandbox environment, after which we demonstrate that improvements in this setting extend to the RL setting as well. We believe the fact that this bonus prevents the diversity collapse phenomenon in RL is a promising signal indicating that deliberate exploration is a practical path toward the discovery of new behaviors beyond sharpening.
>
> Nevertheless, we agree there could be more analysis to see if the model post-trained with RepExp produces behaviors beyond sharpening. To this end, inspired by figure 4 of [1], we have included some additional results where we sample a single response from the base model, the GRPO post-trained model, and the RepExp post-trained model on the full test sets of MATH and GSM8K. We then score all responses under the base model in terms of log likelihood. We plot the resulting histograms [here (for MATH)](https://anonymous.4open.science/r/rep-exp-5594/histogram_math_1.0_hard_1.0_seed_41.pdf) and [here (for GSM8k)](https://anonymous.4open.science/r/rep-exp-5594/histogram_gsm8k_1.0_hard_1.0_seed_41.pdf). We find that in both domains the responses from RepExp tend to be less likely under the base model, quite dramatically so on GSM8K, indicating it’s generating responses that look more novel under the base model and hence not merely performing sharpening. In contrast, notice that standard GRPO exhibits sharpening behavior on MATH, as demonstrated by the movement of probability mass towards the right with respect to the base model.
>
> [1] Karan, Aayush, and Yilun Du. "Reasoning with sampling: Your base model is smarter than you think." arXiv preprint arXiv:2510.14901 (2025).
>
> > In the post-training experiments, the method consistently underperforms GRPO and Unlikeliness for small values of k.
>
> While this is indeed the case, we argue this phenomenon is rather minor since:
> - It’s only for k = 1 that RepExp slightly underperforms GRPO. As soon as k >= 2, it’s on par or better than GRPO in all domains.
> - From a sample-efficiency perspective, it only takes one extra generation to more than make up the gap in pass@1. **However, on the large k side, it often takes hundreds of extra generations from either Unlikeliness or GRPO to match RepExp**. For example, on AIME 2024, GRPO needs 256 samples to reach the same level of performance as RepExp with only 80 samples.

---

> > ### Author Response · Authors · 2025-11-23
> > **Response to Reviewer ppuG (continued)**
> >
> > > All benchmarks employed (GSM8K, MATH, MBPP+, AIME, Game of 24) feature cheap, automatic verifiers where verification cost is negligible relative to model inference. The paper would benefit from experiments in domains where verification is more expensive.
> >
> > We refer the reader to part 2 of our general response, which restate below for convenience:
> > While we agree it would be interesting to investigate our method in domains where verification is more expensive, our paper uses standard datasets to tractably verify the improvement in efficiency by analogy. We believe our experiments on various tasks and model-dataset pairs are extensive. Still, we’d love to see future work evaluate our method in settings like RLHF or protein sequence generation with language models where verification is known to be inherently expensive. As mentioned by reviewer gWc2, experiments of this sort are somewhat out of scope and probably deserve their own full paper.
> >
> > In addition, we’d also like to remind the reviewer that although we believe our inference-time version of RepExp is indeed practical in domains where verification is expensive, the primary purpose of the inference-time selection setting is as a **sandbox to test exploration methods in isolation, without interference from complex RL mechanisms such as optimization and generalization**. The intention is that this sandbox setting can help the design of future exploration methods to be applied during post-training time.
> >
> > > Why did you choose Qwen-2.5-7b-Instruct for the post-training experiments?
> >
> > We chose Qwen-2.5-7b-Instruct since this model performs favorably in the inference time setting. Specifically, in Figure 1, Qwen-2.5-7b-Instruct outperforms random exploration across all tasks. Note that this exactly ties in to the sandbox point we mentioned in the previous answer: **we hypothesize that methods that work well in our novel inference-time selection sandbox setting will perform favorably at post-training time**.
> >
> > We hope our rebuttal clarifies the reviewer’s confusions and relieves some of their concerns.

---

> ### Author Response · Authors · 2025-12-01
> **Additional Results with Expensive Verification (Protein Sequence Generation)**
>
> We wanted to follow up with the reviewer regarding the following concern:
>
> > All benchmarks employed (GSM8K, MATH, MBPP+, AIME, Game of 24) feature cheap, automatic verifiers where verification cost is negligible relative to model inference. The paper would benefit from experiments in domains where verification is more expensive
>
> In addition to our general response part 2, we refer the reviewer to our "Additional Results on Protein Sequence Generation" post, which we restate below for convenience:
>
> We have conducted some preliminary experiments in the domain of protein sequence generation, based on the suggestion from reviewer gWc2. In this domain, verification requires lab work, which is much more time consuming than sampling protein sequences from a generative protein model. Since we can’t actually perform the lab work, we use two proxy metrics as described below.
>
> For our experimental setup, we followed the unconditional generation setup in the ESM3 paper [1] and sampled 2048 protein sequences for fully masked sequence prompts ranging in length from 64 to 916 with increments of size 4 (i.e. {64, 68, ..., 916}). Then, we performed structure prediction for all sampled sequences using ESMFold [2], which returns pLDDT and pTM scores per sequence. We count a sampled sequence as plausible when pLDDT > 0.8 and pTM > 0.8. To get representations for every sequence, we use ESM-C [1] and average the last layer hidden representations. Finally, we perform inference-time selection for every prompt and its 2048 corresponding sampled sequences, using both random selection and RepExp. We find the average samples-to-correct for random to be 240.3 and for RepExp to be 6.7. **This corresponds to a 35.9x verifier efficiency improvement**. We also plot the [corresponding pass@k curves here](https://anonymous.4open.science/r/rep-exp-5594/protein_pass_at_k.pdf), for which **we find up to 56.9x improvements in verifier efficiency**.
>
> We find these results to be very promising and hope they demonstrate to the reviewer that our method can be practical in a domain where verification is genuinely expensive.
>
> **References**
>
> [1] Thomas Hayes et al., Simulating 500 million years of evolution with a language model. Science 387, 850-858 (2025).
>
> [2] Zeming Lin et al., Evolutionary-scale prediction of atomic-level protein structure with a language model. Science 379, 1123-1130 (2023).

---

### Author Response · Authors · 2025-11-23
**General Response to All Reviewers**

We thank all the reviewers for their valuable feedback. In addition to individual responses provided to each reviewer separately, we address a few common concerns below.

**(1) Concern: Does the work discover new behaviors or go beyond sharpening?**

Our work is meant to be **a stepping stone towards moving beyond the sharpening regime** and discovering qualitatively new behaviors. To this end, we first test whether a representation-based approach can guide the search for diverse behaviors in our novel, inference-time selection sandbox environment, after which we demonstrate that improvements in this setting extend to the RL setting as well. We believe the fact that this bonus prevents the diversity collapse phenomenon in RL is a promising signal indicating that deliberate exploration is a practical path toward the discovery of new behaviors beyond sharpening.

Nevertheless, we agree there could be more analysis to see if the model post-trained with RepExp produces behaviors beyond sharpening. To this end, inspired by figure 4 of [1], we have included some additional results where we sample a single response from the base model, the GRPO post-trained model, and the RepExp post-trained model on the full test sets of MATH and GSM8K. We then score all responses under the base model in terms of log likelihood. We plot the resulting histograms [here (for MATH)](https://anonymous.4open.science/r/rep-exp-5594/histogram_math_1.0_hard_1.0_seed_41.pdf) and [here (for GSM8k)](https://anonymous.4open.science/r/rep-exp-5594/histogram_gsm8k_1.0_hard_1.0_seed_41.pdf). We find that in both domains the responses from RepExp tend to be less likely under the base model, quite dramatically so on GSM8K, indicating it’s generating responses that look more novel under the base model and hence not merely performing sharpening. In contrast, notice that standard GRPO exhibits sharpening behavior on MATH, as demonstrated by the movement of probability mass towards the right with respect to the base model.

[1] Karan, Aayush, and Yilun Du. "Reasoning with sampling: Your base model is smarter than you think." arXiv preprint arXiv:2510.14901 (2025).

**(2) Concern: lack of evaluation in a domain where verification is expensive.**

**EDIT**: *In addition to the response below, please refer to our "Additional Results on Protein Sequence Generation" post where we provide very promising preliminary results in the domain of protein sequence generation, where true verification requires lab work and is hence very expensive.*

While we agree it would be interesting to investigate our method in domains where verification is more expensive, our paper uses standard datasets to tractably verify the improvement in efficiency by analogy. We believe our experiments on various tasks and model-dataset pairs are extensive. Still, we’d love to see future work evaluate our method in settings like RLHF or protein sequence generation with language models where verification is known to be inherently expensive. As mentioned by reviewer gWc2, experiments of this sort are somewhat out of scope and probably deserve their own full paper.

In addition, we’d also like to remind the reviewers that although we believe our inference-time version of RepExp is indeed practical in domains where verification is expensive, the primary purpose of the inference-time selection setting is as a **sandbox to test exploration methods in isolation, without interference from complex RL mechanisms such as optimization and generalization**. The intention is that this sandbox setting can help the design of future exploration methods to be applied during post-training time.

**(3) Concern: the proposed method degrades on weaker models.**

We emphasize that **we see the degradation of exploration performance for weaker models as a finding of our work, not a weakness**. One of the main contributions of our work is better understanding when and how representation-based exploration works, and we expect the finding that model strength plays an important role here to inform future work studying representation-based exploration.

We hope the responses to some of the general criticisms above along with the individual responses provided to each reviewer clarify the reviewers’ confusions and relieve some of their concerns.

---

### Author Response · Authors · 2025-11-28
**Additional Results on Protein Sequence Generation**

Reviewers VhJg and ppuG mentioned the paper would benefit from experiments in a practical domain where verification is more expensive. While we believe an extensive study of this merits its own paper, we have conducted some preliminary experiments in the domain of protein sequence generation, based on the suggestion from reviewer gWc2. In this domain, verification requires lab work, which is much more time consuming than sampling protein sequences from a generative protein model. Since we can’t actually perform the lab work, we use two proxy metrics as described below.

For our experimental setup, we followed the unconditional generation setup in the ESM3 paper [1] and sampled 2048 protein sequences for fully masked sequence prompts ranging in length from 64 to 916 with increments of size 4 (i.e. {64, 68, ..., 916}). Then, we performed structure prediction for all sampled sequences using ESMFold [2], which returns pLDDT and pTM scores per sequence. We count a sampled sequence as plausible when pLDDT > 0.8 and pTM > 0.8. To get representations for every sequence, we use ESM-C [1] and average the last layer hidden representations. Finally, we perform inference-time selection for every prompt and its 2048 corresponding sampled sequences, using both random selection and RepExp. We find the average samples-to-correct for random to be 240.3 and for RepExp to be 6.7. **This corresponds to a 35.9x verifier efficiency improvement**. We also plot the [corresponding pass@k curves here](https://anonymous.4open.science/r/rep-exp-5594/protein_pass_at_k.pdf), for which **we find up to 56.9x improvements in verifier efficiency**.

We find these results to be very promising and hope they demonstrate to the reviewers that our method can be practical in a domain where verification is genuinely expensive.

**References**

[1] Thomas Hayes et al., Simulating 500 million years of evolution with a language model. Science 387, 850-858 (2025).

[2] Zeming Lin et al., Evolutionary-scale prediction of atomic-level protein structure with a language model. Science 379, 1123-1130 (2023).

---

### Meta-Review · Area_Chair_d8Ro · 2026-01-06

**Summary:**

This submission receives original ratings of 8, 8, 4, 2. The reviewers consider that the proposed method is a principled approach with extensive demonstration, and the presentation of this work also forms as a strength. The major concerns are about the "novel behavior", the performance at large K. These concerns are valid but do not form the basis to reject the original submission of this work.

**Reviewer Concerns:**

Addressed:
1. Does the proposed method discover new behaviors or go beyond sharpening?
2. Lack of evaluation in a domain where verification is expensive.
3. The proposed method degrades on weaker models.
4. The proposed method performs worse than GRPO and Unlikeliness at small values of k.

Not fully addressed:
1. "novel behavior" empirical evidence
2. the performance at large K.

**Reviewer Scores:**

This submission receives original ratings of 8, 8, 4, 2.

Reviewer VhJg (rating 2) would raise to 4, and reviewer ppug (4) might raise to 6 if further engagements can address the concerns/questions at large k (RL).

---

### Decision · Program_Chairs · 2026-01-26

Accept (Poster)